# Structural insights into selective and dual antagonism of EP2 and EP4 prostaglandin receptors

Yanli Wu [ID][1,5][✉], Heng Zhang [ID][1,5], Jiuyin Xu [ID][2,5], Kai Wu[1], Wen Hu[1], Xinheng He[1], Gaoming Wang[3], Canrong Wu [ID][2][✉] & H Eric Xu [ID][1,2,4][✉]

## Abstract

Prostaglandin E2 (PGE2) signaling through EP2 and EP4 receptors is crucial in regulating inflammation, pain, and cancer progression. While selective and dual antagonists for these receptors hold therapeutic potential, their binding mechanisms and selectivity have remained unclear. In this study, we present cryo-electron microscopy (cryo-EM) structures of human EP2 and EP4 receptors in complex with selective antagonists PF-04418948 and grapiprant, as well as with the dual antagonist TG6-129. These structures reveal distinct binding pockets and interaction networks that dictate antagonist selectivity and efficacy. Notably, TG6-129 displays a novel binding mode, engaging deeply with EP2 while interacting more superficially with EP4 in a two-warhead manner. Furthermore, comparisons of active and inactive receptor structures elucidate the mechanisms underlying EP2 activation and antagonism. Overall, these findings provide a structural framework for understanding prostanoid receptor pharmacology and offer valuable insights for the rational design of improved selective and dual antagonists targeting EP2 and EP4 receptors.

**Keywords** Cryo-EM; EP2; EP4; Selective Antagonist; Dual Antagonist
**Subject Categories** Pharmacology & Drug Discovery; Signal Transduction; Structural Biology

## Introduction

Prostaglandin $E_2$ (PGE$_2$) is a key lipid mediator that regulates diverse physiological processes, including inflammation, pain sensation, and cancer progression (Tsuge et al, 2019). PGE$_2$ exerts its effects through four G protein-coupled receptors (GPCRs): EP1, EP2, EP3, and EP4, coupled with different signaling pathways: EP1 induces intracellular Ca$^{2+}$ influx through G$_q$ protein, EP3 reduces cyclic adenosine monophosphate (cAMP) by inhibiting adenylate cyclase through G$_i$, whereas EP2 and EP4 increase the production

cAMP by activating adenylyl cyclase through G$_s$ (Regan, 2003; Tsuge et al, 2019). Although EP2 and EP4 receptors act redundantly in some processes, they exhibit distinct tissue distribution patterns and can activate different downstream effectors, resulting in unique physiological roles (Santiso et al, 2024). EP2 differs from all other PG receptors in that EP2 does not undergo homologous desensitization upon PGE$_2$ binding, thus acting over prolonged periods (Malty et al, 2016). EP2-activated G$_s$ also activates the β-catenin pathway independently of cAMP. Whereas, EP4 can activate phosphatidylinositol 3-kinase (PI3K) pathway cAMP-independently (Regan, 2003; Santiso et al, 2024).

PGE$_2$ promotes an immunosuppressed tumor microenvironment (TME) by affecting immune cells mainly through EP2 and EP4 receptors (Finetti et al, 2020). The binding of PGE$_2$ to EP2 activates a positive feedback regulation of cyclooxygenase 2 (COX-2), a key enzyme for PGE$_2$ synthesis, leading to even higher levels of PGE$_2$ in the TME and amplification of PGE$_2$ signaling (Obermajer et al, 2011). Recent investigations have suggested that PGE$_2$-EP2/EP4 signaling in TME boosts inflammation and angiogenesis through NF-κB, and elicits immunosuppression through Tregs recruitment and activation (Thumkeo et al, 2022). In addition, PGE$_2$-EP2/EP4 axis also inhibits NK activity in TME and restricts CD8$^+$ T cell-mediated anticancer immunity and thus promotes cancer immune escape (Bonavita et al, 2020; Lacher et al, 2024; Morotti et al, 2024). Therefore, the PGE$_2$-EP2/EP4 signaling pathway serves as a key regulatory node connecting active inflammation and immune suppression in TME, and can be targeted by antagonists for cancer treatment.

These recent findings in cancer immunology have reinforced the importance of EP2 and EP4 as therapeutic targets, reigniting interest in developing antagonists for these receptors. PF-04418948, introduced by Pfizer in 2011, is a potent and selective EP2 antagonist with potential applications in inflammation-associated diseases, including endometriosis (af Forselles et al, 2011; Birrell et al, 2013). Similarly, grapiprant (CJ-023423), a potent specific antagonist of EP4, has been approved for the control of pain and inflammation associated with osteoarthritis in dogs (Nakao et al, 2007; Sartini and Giorgi, 2021). Grapiprant is currently undergoing clinical trials for human applications in inflammation-associated diseases, cancer, and cancer-related pain. While selective antagonists have demonstrated promising effects, the

[1]State Key Laboratory of Drug Research, Shanghai Institute of Materia Medica, Chinese Academy of Sciences, 201203 Shanghai, China. [2]Research Center for Medicinal Structural Biology, National Research Center for Translational Medicine at Shanghai, State Key Laboratory of Medical Genomics, Ruijin Hospital, Shanghai Jiao Tong University School of Medicine, 200025 Shanghai, China. [3]Department of Biliary-Pancreatic Surgery, Renji Hospital, Shanghai Jiao Tong University School of Medicine, 200025 Shanghai, China. [4]University of Chinese Academy of Sciences, Beijing 100049, China. [5]These authors contributed equally: Yanli Wu, Heng Zhang, Jiuyin Xu. [✉]E-mail: wuyanli@simm.ac.cn; wcr13215@rjh.com.cn; eric.xu@simm.ac.cn

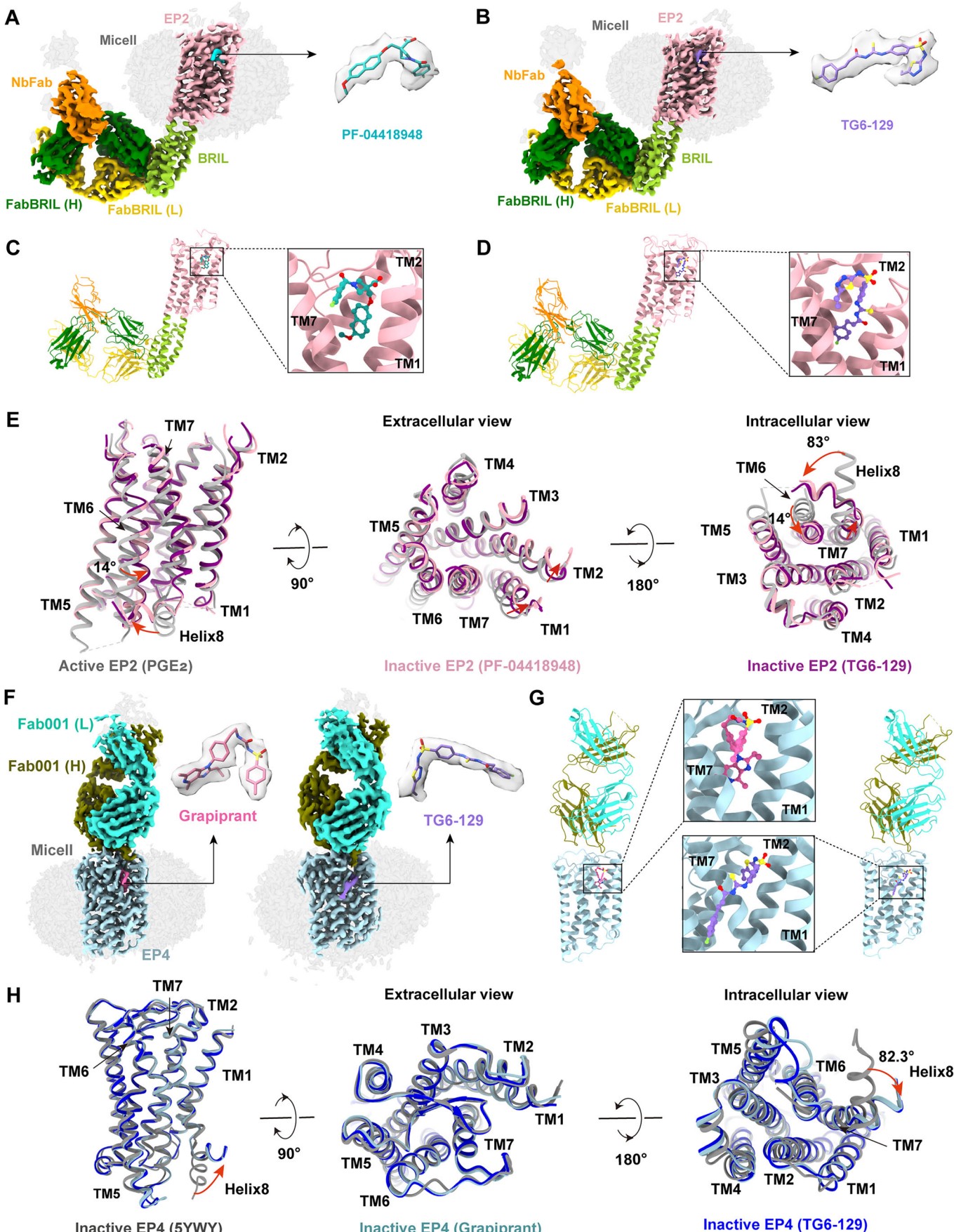

**Figure 1.  Overall structures of inactive EP2 and EP4 complexes.**

(A, B) Cryo-EM density map of EP2-PF-04418948 complex (**A**) and EP2-TG6-129 complex (**B**). PF-04418948 and TG6-129 are shown as sticks in a zoomed-in view. EP2 in light pink, BRIL in yellow green, FabBRIL (H) in green, FabBRIL (L) in gold, NbFab in dark orange. (**C, D**) Cartoon models of EP2-PF-04418948 complex (**C**) and EP2-TG6-129 complex (**D**). PF-04418948 and TG6-129 are shown as ball sticks, with the binding pocket is shown in a zoomed-in view. (**E**) Side view (left), extracellular view (middle), and intracellular view (right) of structural comparisons between the inactive EP2 and the active EP2 structure (PDB ID: 7CX2). Active EP2 (PGE$_2$), gray; Inactive EP2 (PF-04418948), light pink; Inactive EP2 (TG6-129), purple. The conformational changes are shown by red arrows. (**F**) Cryo-EM density map of EP4-grapiprant complex (left) and EP4-TG6-129 complex (right). Grapiprant and TG6-129 are shown as sticks in a zoomed-in view. EP4 in light blue, Fab001 (H) in olive, Fab001 (L) in turquoise. (**G**) Cartoon models of EP4-grapiprant complex (left) and EP4-TG6-129 complex (right). Grapiprant and TG6-129 are shown as ball sticks, with the binding pocket is shown in a zoomed-in view. (**H**) Side view (left), extracellular view (middle), and intracellular view (right) of structural comparisons between the inactive EP4 complexes. Inactive EP4 (PDB ID: 5YWY), dark gray; Inactive EP4 (grapiprant), light blue; Inactive EP4 (TG6-129), blue. The conformational changes are shown by red arrows.

concept of dual EP2/EP4 inhibition has gained traction due to the potential for enhanced efficacy and broader treatment options through counteracting the effects of PGE$_2$ more comprehensively. TG6-129, as a dual-targeting antagonist with preferential binding to EP2, represents a step towards this goal, exhibiting antagonism towards both EP2 and EP4, albeit with differing potencies (Ganesh et al, 2013).

Several structural pharmacological studies have provided valuable insights into ligand recognition by prostanoid receptors. Active form structures have been elucidated for EP2 (Qu et al, 2021), EP3 (Morimoto et al, 2019), EP4 (Huang et al, 2023; Nojima et al, 2021), prostaglandin F2α receptor FP (Wu et al, 2023), thromboxane receptor TP (Li et al, 2024), prostacyclin receptor IP (Wang et al, 2024), and prostaglandin D2 receptors DP2 (Xu et al, 2024). However, inactive structures are currently limited to EP4 (Toyoda et al, 2019), TP (Fan et al, 2019), and prostaglandin D2 receptors DP2 (Wang et al, 2018). Despite these advancements, the structural basis for the selective and dual antagonism of EP2 and EP4 receptors remained elusive.

Understanding the molecular mechanisms underlying ligand binding and selectivity is crucial for the rational design of improved antagonists with enhanced potency, selectivity, or dual-targeting capabilities. To address this knowledge gap, our study presents cryo-electron microscopy (cryo-EM) structures of human EP2 and EP4 receptors in complex with their respective selective antagonists, PF-04418948 and grapiprant, as well as the dual antagonist TG6-129. These structures provide important insights into the binding modes of these antagonists and reveal key structural features that govern their selectivity and efficacy, and also elucidate the mechanisms underlying EP2 activation and antagonism. Our findings highlight distinct binding pockets and interaction networks in EP2 and EP4 receptors, elucidating the molecular basis for selective antagonism. Furthermore, we uncover a novel binding mode for the dual antagonist TG6-129. These structural insights will facilitate the rational design of next-generation selective and dual antagonists targeting EP2 and EP4 receptors, potentially leading to more effective therapeutic strategies for inflammation-associated diseases and cancer.

## Results

### Overall structures of inactive EP2 and EP4 complexes

To examine the antagonism profiles of the three distinct ligands, PF-04418948, grapiprant, and TG6-129, on EP2 and EP4, respectively, we performed the cAMP accumulation assay to investigate their antagonistic activity upon PGE$_2$ stimulation. As

reported, it revealed that PF-04418948 and grapiprant are selective antagonists of EP2 and EP4, respectively, while TG6-129 is a dual antagonist that preferentially antagonizes EP2 (Appendix Fig. S1A). To explore the binding modes of selective and dual antagonists for EP2 and EP4 receptors, we employed genetic engineering strategies and antibodies to obtain stable EP2 and EP4 proteins for cryo-EM studies.

For EP2, the third intracellular loop (ICL3) between TM5 and TM6 was replaced with BRIL, which was further stabilized by engagement with an anti-BRIL Fab (FabBRIL) (Mukherjee et al, 2020) and an anti-Fab nanobody (NbFab) (Ereno-Orbea et al, 2018), as such a method has been used to solve a number of inactive GPCR structures (Lees et al, 2023; Tsutsumi et al, 2020). This BRIL-fusion EP2 was purified and subsequently incubated with either PF-04418948 (selective antagonist) or TG6-129 (dual antagonist), along with FabBRIL and NbFab, to produce highly homogeneous complex samples for structural studies (Appendix Fig. S1B and Appendix Table S1). The structures of EP2-BRIL in complex with PF-04418948 or TG6-129 were determined by cryo-EM at overall resolutions of 3.50 Å and 3.28 Å, respectively (Fig. 1A–D; Appendix Figs. S2 and S3). The EM density maps provided a reliable model for the EP2 structure containing residues 12–328, except for the intracellular loop 1 (ICL1) (residues 52–63) and ICL3 (residues 225–256, replaced by BRIL). The density maps were also clear for the two antagonists and most residues of the anti-BRIL Fab and NbFab (Fig. 1A–D).

For EP4, we introduced truncation and mutation strategies as described in a previous study (Toyoda et al, 2019). This modified EP4 was purified and then incubated with Fab001, an antibody that binds to allosteric sites and promotes the stability of inactive EP4 complex (Toyoda et al, 2019), in the presence of either grapiprant (selective antagonist) or TG6-129 (dual antagonist). The cryo-EM structures of EP4-Fab001 in complex with grapiprant or TG6-129 were determined at overall resolutions of 2.65 Å and 2.92 Å, respectively (Fig. 1F,G; Appendix Figs. S4 and S5).

Upon obtaining the structures, we compared the antagonist-bound inactive EP2 structures with the PGE$_2$-bound active EP2 structure (PDB code: 7CX2) (Qu et al, 2021). Significant conformational changes were observed between the active and inactive states (Fig. 1E). The inactive EP2 exhibited a gap between TM1 and TM7 on the extracellular side, which was closed in the active state. TM6 of the inactive state EP2 retracted towards the receptor on the middle and intracellular sides, with an angular difference of ~14°, while TM7 shifted towards the outside of the receptor on the intracellular side. In addition, helix 8 of the inactive state EP2 extended in a different direction compared to the active state, with an angular difference of ~83°. These conformational

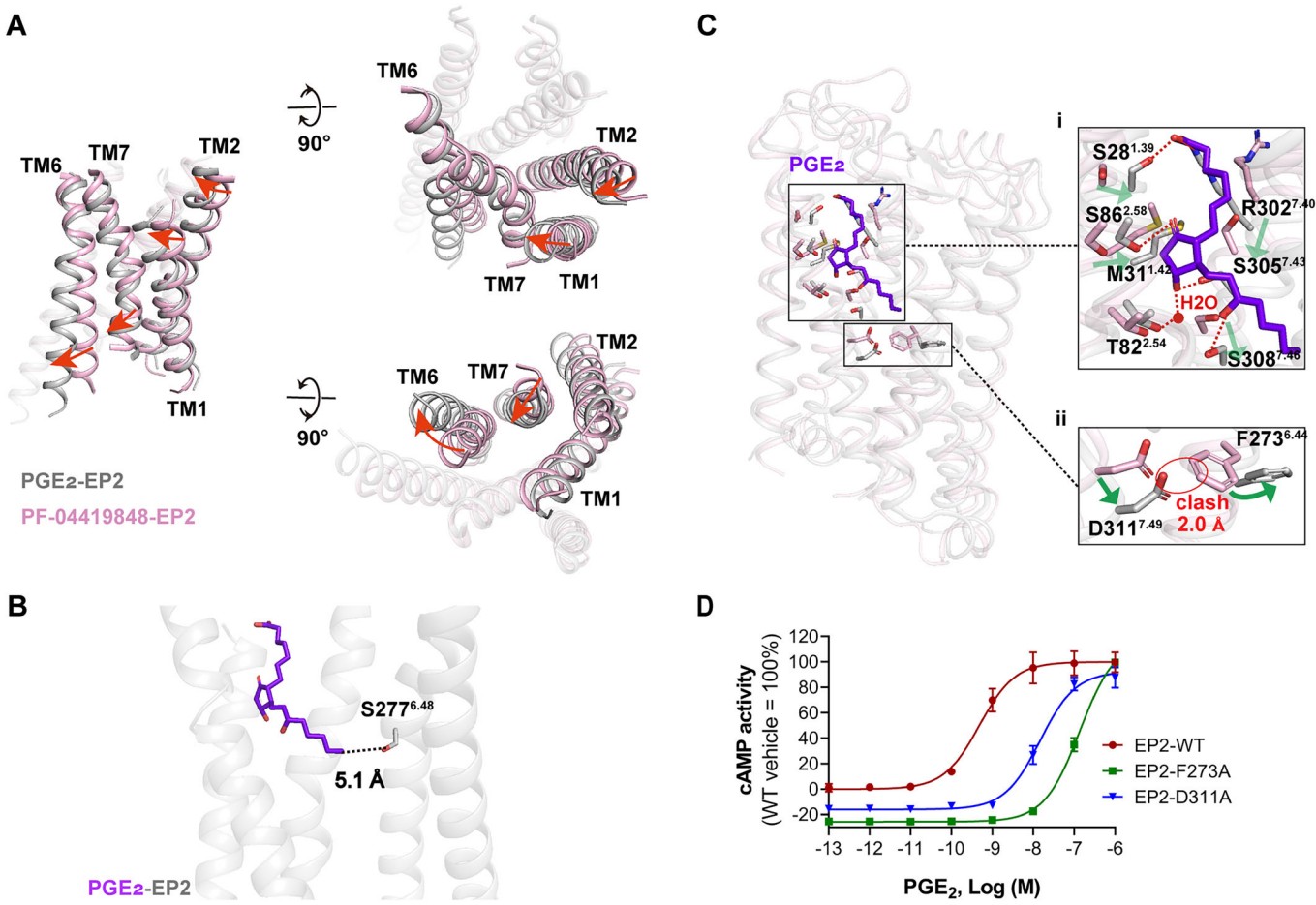

**Figure 2. Activation of EP2.**

(A) Structure comparison of PGE₂-EP2 (gray) active and PF-04418948-EP2 (light pink) inactive structures. The conformational changes are shown by red arrows. (B) Superposition of PGE₂-bound EP2, presenting its steric distance to S277$^{6.48}$. (C) Structure superposition of the active (gray) and inactive (light pink) structures of EP2. PGE₂-induced conformational changes of residues in the pocket (i) and in the TM7 and TM6 (ii) are depicted in a zoomed-in view. The hydrogen bond is shown as a red dashed line. The red circle indicates the clash between the atoms. The conformational changes are shown by green arrows. (D) cAMP responses of key mutants in EP2 activation. For each EP2 mutant, cAMP activity (%) is compared with EP2-WT. Data are presented as mean ± S.E.M. of three independent experiments with three technical replicates, respectively. Source data are available online for this figure.

transitions of TM1, TM6, and TM7 from the active to the inactive state in EP2 are typical of other lipid receptors.

Similarly, we compared the overall structures of the inactive EP4 receptor with grapiprant/TG6-129 to the previously determined antagonist ONO-AE3-208-bound EP4 (PDB code: 5YWY) (Toyoda et al, 2019). As expected, their overall structures were highly similar, with root mean square deviation (RMSD) values of 0.817 and 0.797 Å, respectively. A notable structural difference was observed in helix 8, which in the cryo-EM structures was almost perpendicular to that in the crystal structure, with a rotation of 82.3° (Fig. 1H). This difference in helix 8 may be caused by crystal packing, as helix 8 was involved in crystal contacts in the crystal structure (Toyoda et al, 2019).

## Activation of EP2

The molecular mechanism by which PGE₂ activates EP2 remains incompletely understood due to the lack of an inactive

EP2 structure. To investigate the activation mechanism of EP2, we compared the structures of inactive EP2 and PGE₂-bound EP2, revealing a series of structural rearrangements linked to activation (Fig. 2A). Notably, the presence of a smaller serine (S) at residue 6.48, rather than tryptophan (W), enlarges the distance of this residue to the bound ligand, suggesting that the ligand does not directly interact with TM6 (Fig. 2B), as seen in many class A GPCR activation.

Instead, our structural analysis reveals a significant clash between TM7 in active EP2 and TM6 in inactive EP2, indicating that the rearrangement of TM7 is crucial for receptor activation. In the ligand-binding pocket, we observed that the upper region of the active conformational pocket is more constricted, with TM1 and TM2 shifting inward. This inward movement is primarily driven by strong interactions between PGE₂ and residues in TM1 and TM2 (Fig. 2C, i).

As TM1 and TM2 rearrange, they cause TM7 to shift downward, which in turn pushes residue F273 in TM6 through interactions

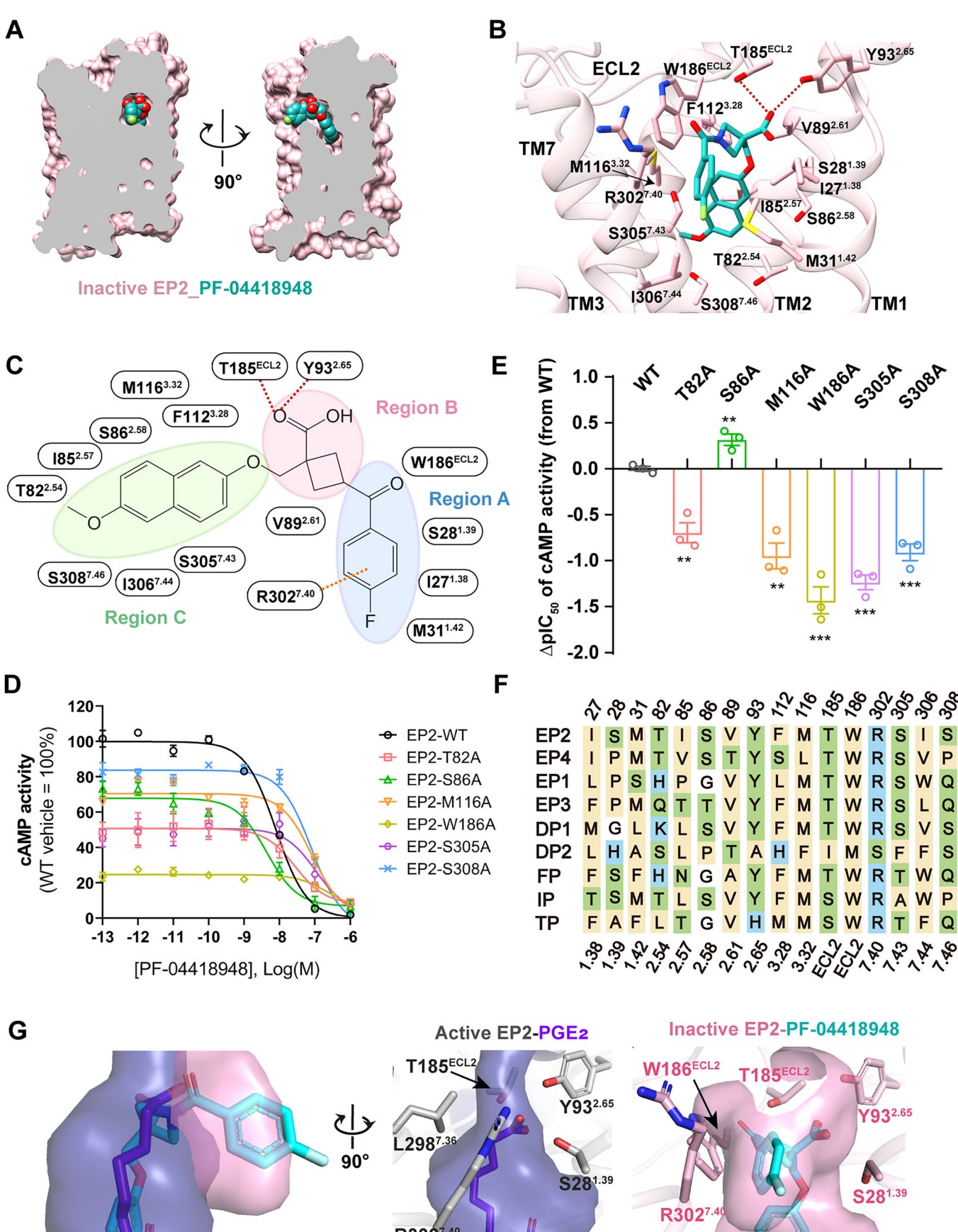

**Figure 3.  Selective inhibition of EP2.**

(A) Vertical cross-section of PF-04418948 binding pocket in EP2. (B, C) 3D presentation (B) and 2D format of region division (C) for corresponding interactions that contribute to PF-04418948 binding in EP2. The hydrogen bond is depicted as a red dashed line. The cation−π interaction is shown as an orange dashed line. (D, E) cAMP response (D) and $\Delta pIC_{50}$ (E) of PF-04418948 in $PGE_2$-induced EP2 mutants. $\Delta pIC_{50} = pIC_{50}$ of PF-04418948 to specific mutant - $pIC_{50}$ of PF-04418948 to WT. Data are presented as mean ± S.E.M. of three independent experiments with three technical replicates each. Significance was determined with a two-sided unpaired t test. Compared with EP2 WT, the P values in panel E are 0.0035, 0.0091, 0.0026, 0.0006, 0.0001, and 0.0006 for EP2 T82A, S86A, M116A, W186A, S305A, and S308A mutants, respectively. **$P \le 0.01$, ***$P \le 0.001$. (F) sequence alignment of key residues participated in ligand binding for nine prostanoid receptors. Hydrophobic residues are in yellow, polar charged residues in blue, and polar uncharged residues in green. (G) Comparison of the overall shape (left) and the specific residues around the top of the binding pockets in $PGE_2$-bound active (middle, blue) and PF-04418948-bound inactive (right, light pink) EP2. Source data are available online for this figure.

with D311, facilitating the outward movement of TM6 (Fig. 2C, ii). Notably, mutations of D311, or F273 to alanine significantly reduce the potency of $PGE_2$ in activating EP2 (Fig. 2D; Appendix Fig. S6A,B and Appendix Table S2).

## Selective inhibition of EP2

PF-04418948 is a highly selective antagonist for EP2 with a $K_B$ value of 1.8 nM against $PGE_2$-induced cAMP signaling in GloSensor cAMP assay (af Forselles et al, 2011). To understand the structural basis of the antagonistic selectivity of PF-04418948 on EP2, and to facilitate the future design of selective or dual antagonists, we determined the cryo-EM structure of the PF-04418948-EP2 complex. This structure enabled the precise localization of PF-04418948 within the orthosteric binding pocket of EP2.

The inactive structure of EP2 bound to PF-04418948 revealed both common characteristics and distinct features compared to the active structure of $PGE_2$-bound EP2 (PDB code: 7CX2) (Qu et al, 2021). The binding pocket of the $PGE_2$-EP2 complex contains a solvent-accessible channel defined by Y93^2.65 and P183^ECL2 on the extracellular side, and $PGE_2$ assumes a deeper, more extended conformation (Fig. EV1A). Similar to that of the inactive EP4 structure, the surface representation showed that the binding pocket of the PF-04418948-EP2 complex is closed to the extracellular region but open to the lipid bilayer between TM1 and TM7 (Figs. 3A and EV1A).

PF-04418948 binds in a pocket formed by the side chains of ECL2 and TMs 1-3 and 7 (Figs. 3B and EV1B). The antagonist assumes an L-shaped conformation within its ligand-binding pocket and is mainly composed of three regions: A, B, and C (Fig. 3C). Region A occupies the entrance of the binding pocket and is primarily hydrophobic. It mainly forms hydrophobic interactions with residues I27^1.38, M31^1.42, and W186^ECL2. In addition, the 3-fluorophenyl ring in region A forms a cation−π interaction stack with the side chain of R302^7.40. Mutation of W186^ECL2 to alanine reduced the potency of PF-04418948 on antagonize EP2, suggesting the critical role of W186^ECL2 in PF-04418948 recognition (Figs. 3D,E and EV1C−E; Appendix Table S3).

Region B, containing the carboxyl group of PF-04418948, is located at the top of the polar region of the binding pocket. The carboxyl group interacts through hydrogen bonds with Y93^2.65 and T185^ECL2, key residues highly conserved across the prostanoid receptor family (Fig. 3F). Region C is composed of a hydrophobic moiety extending into a pocket formed by TM2, TM3, and TM7 of EP2. It forms extensive interactions with residues T82^2.54, I85^2.57, S86^2.58, V89^2.61 in TM2, F112^3.28, M116^3.32 in TM3, and S305^7.43, I306^7.44,

S308^7.46 in TM7. Consistently, mutations such as T82^2.54A, M116^3.32A, S305^7.43A, or S308^7.46A in EP2 significantly reduced the potency of PF-04418948 in antagonizing EP2 (Figs. 3D,E and EV1C−E; Appendix Table S3). Unexpectedly, S86^2.58A mutation enhanced PF-04418948 antagonistic activity against $PGE_2$-stimulated EP2 receptor (Fig. 3D), but decreased its antagonistic potency against constitutively active EP2 receptor (Fig. EV1E). The different change may be due to the weakened activation potency of $PGE_2$ on mutant EP2, which favors the ligand antagonism.

Structural comparison of the $PGE_2$-bound active EP2 structure with the PF-04418948-bound inactive EP2 structure provides an opportunity to explore the antagonism mechanism for EP2. In the case of active state EP2, the top of the binding pocket of $PGE_2$ is surround by the side chains of several residues from TM1, TM2, TM7 and ECL2, forming a narrow and closed binding cavity (Fig. 3G). in contrast, PF-04418948 is positioned shallower, with Region A and B occupying the top of the binding pocket, forming an enlarged and protruding pocket cavity that widens the gap between TM1 and TM7 (Fig. 3G). In addition, PF-04418948 mainly forms weak van der Waals interaction, instead of strong polar interaction, with residues S86^2.58, T82^2.54 in TM2, and S305^7.43, S308^7.46 in TM7 (Fig. 3B,C), which are not sufficient to drive a downward shift of TM7, blocking the subsequent rearrangement of TM7 and TM6, thus achieving an antagonistic conformation.

## Selective inhibition of EP4

To explore the structural basis of EP4-selective antagonism, we determined the cryo-EM structure of the grapiprant-EP4 complex. Grapiprant is a selective antagonist against EP4 with the $K_i$ value of 13 nM in [³H]-$PGE_2$ binding assay (Nakao et al, 2007). The structure was obtained at a resolution of 2.65 Å, providing excellent local resolution for both the receptor and the assignment of grapiprant.

Similar to the reported antagonist ONO-AE3-208-bound inactive structure of EP4 (Toyoda et al, 2019), the grapiprant-bound EP4 structure displays a gap between TM1 and TM7 within the membrane, suggesting this gap is likely the entrance for the ligand from the membrane (Fig. 4A). Grapiprant binds in a pocket formed by the side chains of ECL2 and TMs 1–3 and 7, assuming an L-shaped conformation via a three-part mode (Fig. 4B,C).

In Region A, grapiprant primarily interacts with M27^1.42 and V320^7.44 through hydrophobic interactions. At the top of the polar region of the binding pocket, the carboxyl group and the sulfonyl group of grapiprant in Region B interact through hydrogen bonds with the guanidinium group of R316^7.40 and the hydroxyl

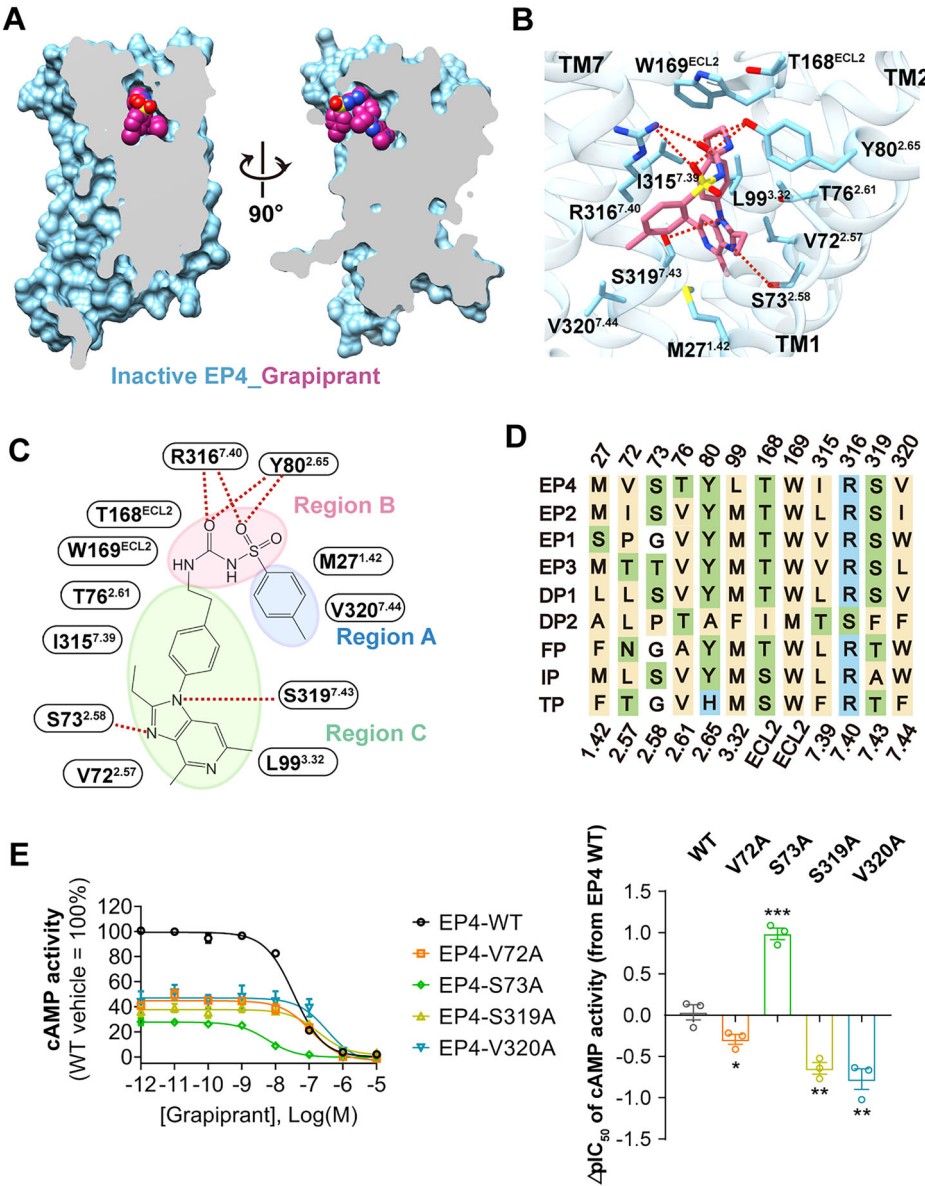

**Figure 4. Selective inhibition of EP4.**

(A) Vertical cross-section of grapiprant binding pocket in EP4. (B, C) 3D presentation (B) and 2D format of region division (C) for corresponding interactions that contribute to grapiprant binding in EP4. The hydrogen bond is depicted as a red dashed line. (D) sequence alignment of key residues participated in ligand binding for nine prostanoid receptors. Hydrophobic residues are in yellow, polar charged residues in blue, and polar uncharged residues in green. (E) cAMP response (left) and $\Delta pIC_{50}$ (right) of grapiprant in PGE₂-induced EP4 mutants. $\Delta pIC_{50} = pIC_{50}$ of grapiprant to specific mutant - $pIC_{50}$ of grapiprant to WT. Data are presented as mean ± S.E.M. of three independent experiments with three technical replicates, respectively. Significance was determined with a two-sided unpaired $t$ test. Compared with EP4 WT, the $P$ values in the right panel are 0.0383, 0.0001, 0.0013, and 0.0013 for EP4 V72A, S73A, S319A, and V320A mutants, respectively. $*P \leq 0.05$, $**P \leq 0.01$, $***P \leq 0.001$. Source data are available online for this figure.

group of Y80²·⁶⁵, which are highly conserved among the prostanoid receptor family (Fig. 4D). In this region, grapiprant forms a tight interaction network between the ligand and the binding site, also involving a cation−π interaction with the guanidinium groups of R316⁷·⁴⁰.

In Region C, grapiprant forms extensive hydrophobic interactions with residues that are highly conserved among prostanoid receptors, including T168^{ECL2}, W169^{ECL2}, as well as nonconserved residues V72²·⁵⁷, T76²·⁶¹, L99³·³², and I315⁷·³⁹. In addition, the residues S73²·⁵⁸

and S319⁷·⁴³ interact through hydrogen bonds with the imidazole ring of grapiprant, contributing to a more extended binding cavity than the ONO-AE3-208-binding pocket (Fig. EV2A,B). Among these non-conserved residues, V72²·⁵⁷, T76²·⁶¹, L99³·³², I315⁷·³⁹, and V320⁷·⁴⁴ surrounding grapiprant are unique to EP4, which contribute in selectivity of grapiprant. Consistently, mutations such as V72²·⁵⁷, S319⁷·⁴³A, and V320⁷·⁴⁴A reduced the potency of grapiprant on antagonizing PGE₂-induced and constitutively active EP4-cAMP signaling (Figs. 4E and EV2C,D). Unexpectedly, S73²·⁵⁸A mutation

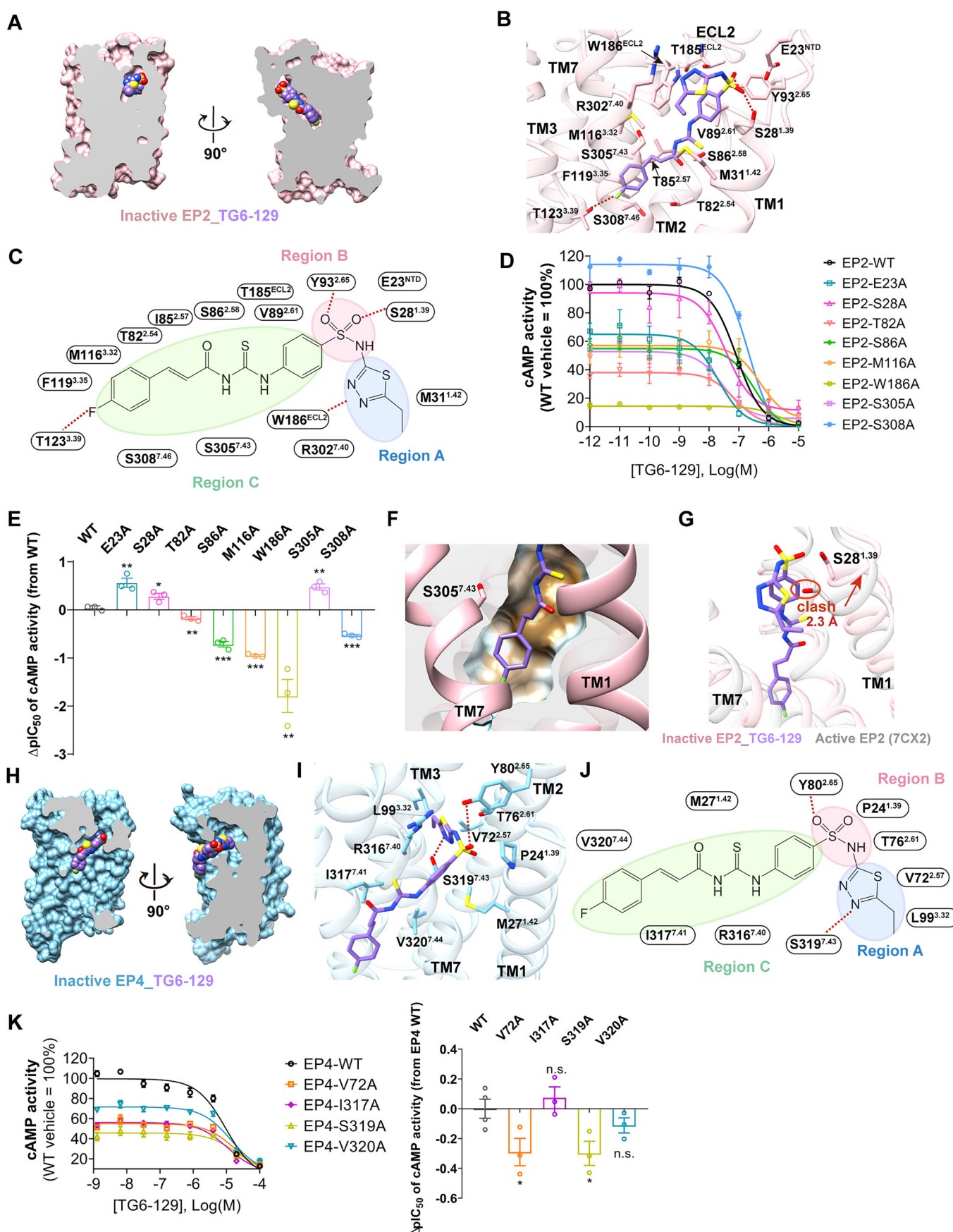

Figure 5. Dual inhibition of EP2 and EP4.

(A) Vertical cross-section of TG6-129 binding pocket in EP2. (B, C) 3D presentation (B) and 2D format of region division (C) for corresponding interactions that contribute to TG6-129 binding in EP2. The hydrogen bond is depicted as a red dashed line. (D, E) cAMP response (D) and $\Delta pIC_{50}$ (E) of TG6-129 in $PGE_2$-induced EP2 mutants. $\Delta pIC_{50} = pIC_{50}$ of TG6-129 to specific mutant - $pIC_{50}$ of TG6-129 to WT. Compared with EP2 WT, the $P$ values in panel E are 0.0063, 0.0234, 0.0064, 0.0003, 0.0000, 0.0060, 0.0034, and 0.0001 for EP2 E23A, S28A, T82A, S86A, M116A, W186A, S305A, and S308A mutants, respectively. (F) TG6-129 and S305[7.43] in the binding pocket of the TG6-129-EP2 complex structure. (G) Comparison between active EP2 and TG6-129-bound EP2. The red circle indicates the clash between the atoms. The conformational changes are shown by red arrows. (H) Vertical cross-section of TG6-129 binding pocket in EP4. (I, J) 3D presentation (I) and 2D format of region division (J) for corresponding interactions that contribute to TG6-129 binding in EP4. The hydrogen bond is depicted as a red dashed line. (K) cAMP response (left) and $\Delta pIC_{50}$ (right) of TG6-129 in $PGE_2$-induced EP4 mutants. $\Delta pIC_{50} = pIC_{50}$ of TG6-129 to specific mutant - $pIC_{50}$ of TG6-129 to WT. Compared with EP4 WT, the $P$ values are 0.0433, 0.4834, 0.0322, and 0.2566 for EP4 V72A, I317A, S319A, and V320A mutants, respectively. Data information: Data are presented as mean ± S.E.M. of three independent experiments with three technical replicates, respectively. Significance was determined with a two-sided unpaired $t$ test; *$P \leq 0.05$, **$P \leq 0.01$, ***$P \leq 0.001$, n.s. $P > 0.05$. Source data are available online for this figure.

enhanced the antagonism of grapiprant against $PGE_2$-stimulated EP4 receptor (Fig. 4E), but hardly influenced antagonistic potency against constitutively active EP4 receptor (Fig. EV2D). The different change may be due to the weakened activation potency of $PGE_2$ on mutant EP4.

## Dual inhibition of EP2 and EP4

$PGE_2$ promotes tumorigenesis, metastasis, and immune suppression through both EP2 and EP4 receptors (Finetti et al, 2020; Santiso et al, 2024). While selective EP2 or EP4 antagonists have shown promising anticancer effects, dual inhibition of both receptors has the potential to counteract the effects of $PGE_2$ more comprehensively, potentially resulting in increased efficacy and broader treatment options. To explore this approach, we focused on TG6-129, a dual-targeting antagonist with preferential binding to EP2 (Ganesh et al, 2013). TG6-129 exhibits potent antagonism towards EP2 (Kd: 8.8 nM) and weaker antagonism towards EP4 (Kd: 3.9 μM) (Ganesh et al, 2013). To elucidate the structural basis of TG6-129 with dual binding capability, we obtained cryo-EM structures of TG6-129 bound to both EP2 and EP4 receptors.

In the EP2 complex, TG6-129 binds in a manner similar to PF-04418948, occupying a pocket formed by ECL2 and TMs 1-3 and 7. The antagonist likely enters through the gap between TM1 and TM7 from the lipid bilayer (Fig. 5A). The binding mode of TG6-129 in EP2 can be divided into three regions, mirroring the PF-04418948 binding pattern (Figs. 5B,C and EV3A).

In Region A, the 1,3,4-thiadiazole ring of TG6-129 forms crucial hydrogen bonds with W186[ECL2] and R302[7.40]. The importance of W186[ECL2] is underscored by the significant reduction in the antagonistic potency of TG6-129 when this residue is mutated to alanine (Figs. 5D,E and EV3B). Region B, located at the top of the polar binding pocket, involves the sulfonyl group of TG6-129 forming hydrogen bonds with S28[1.39] and Y93[2.65]. Interestingly, the nearby acidic residue E23[NTD] exerts a repulsive effect, as evidenced by the increased potency of TG6-129 when E23[NTD] is mutated to alanine.

Region C of TG6-129 extends deep into the EP2 binding pocket, interacting with residues from TM2 (T82[2.54], I85[2.57], S86[2.58], V89[2.61]), TM3 (M116[3.32], F119[3.35]), and TM7 (S305[7.43], S308[7.46]). Mutations of key residues in this region, such as T82[2.54]A, S86[2.58]A and M116[3.32]A, significantly reduced the antagonistic potency of TG6-129 on EP2 (Figs. 5D,E and EV3B). An intriguing observation is the packing of the polar residue S305[7.43] with the hydrophobic

part of the Region C in TG6-129 (Fig. 5F). Substituting S305[7.43] with alanine enhanced the antagonistic potency of TG6-129, likely due to improved hydrophobic packing. S308[7.46]A mutation hardly affects the antagonistic potency of TG6-129 against constitutive activation, while significantly reducing antagonism of $PGE_2$-stimulated activity (Figs. 5E and EV3B), indicating the importance of S308[7.46] in the competitive binding of the orthosteric pocket.

Structural comparisons between the inactive TG6-129-bound EP2 and the active $PGE_2$-bound EP2 (PDB code: 7CX2) revealed conformational shifts that create clashes between TG6-129 and key residues in TM1 (Fig. 5G), consistent with the potential mechanisms of antagonism. This is further supported by the increase in the potency of TG6-129 observed with the S28[1.39]A mutation (Figs. 5E and EV3B).

In contrast to its deep binding in EP2, TG6-129 exhibits a markedly different binding mode in EP4. Most of the molecule extends outside the orthosteric binding pocket, binding shallowly at the top via Region A (Figs. 5H–J and EV3C). This shallow binding is in stark contrast to grapiprant, which integrates deeply into the EP4 pocket via its Region C.

In Region A, TG6-129 primarily forms hydrogen bonds with S319[7.43] and hydrophobic interactions with V72[2.57], T76[2.61], and L99[3.32]. Region B involves the sulfonyl group of TG6-129 hydrogen bonding with the highly conserved Y80[2.65], a residue crucial for ligand binding across the prostanoid receptor family. Region C of TG6-129 binds outside the pocket, engaging in hydrophobic interactions with M27[1.42] in TM1 and R316[7.40], I317[7.41], and V320[7.44] in TM7 (Figs. 5H–J and EV3C). Notably, several interacting residues (V72[2.57], T76[2.61], L99[3.32] and V320[7.44]) are nonconserved, with some being unique to EP4. Mutations of V72[2.57]A and S319[7.43]A significantly reduced the antagonistic potency of TG6-129 on EP4 (Figs. 5K and EV3D).

Structural comparisons between TG6-129-bound EP2 and TG6-129-bound EP4 revealed that compared with EP2, TM2 of EP4 in the binding pocket, shows a sharp inward movement, accompanied by slight inward movement of TM7 and outward movement of TM5, and therefore the binding position of TG6-129 in EP4 is shifted towards TM7 as a whole compared to EP2 (Fig. EV3E). In EP2, TG6-129 deeply interacts with TM2 and TM3 inside the pocket via Region C, and interacts with TM1, TM7, and ECL2 at the top of the binding pocket. while in EP4, TG6-129 mainly interacts with TM7 on the outside of the pocket via Region C (Fig. EV3Ea–c). This distinct binding mode highlights the structural differences between EP2 and EP4 that contribute to the dual antagonistic activity of TG6-129.

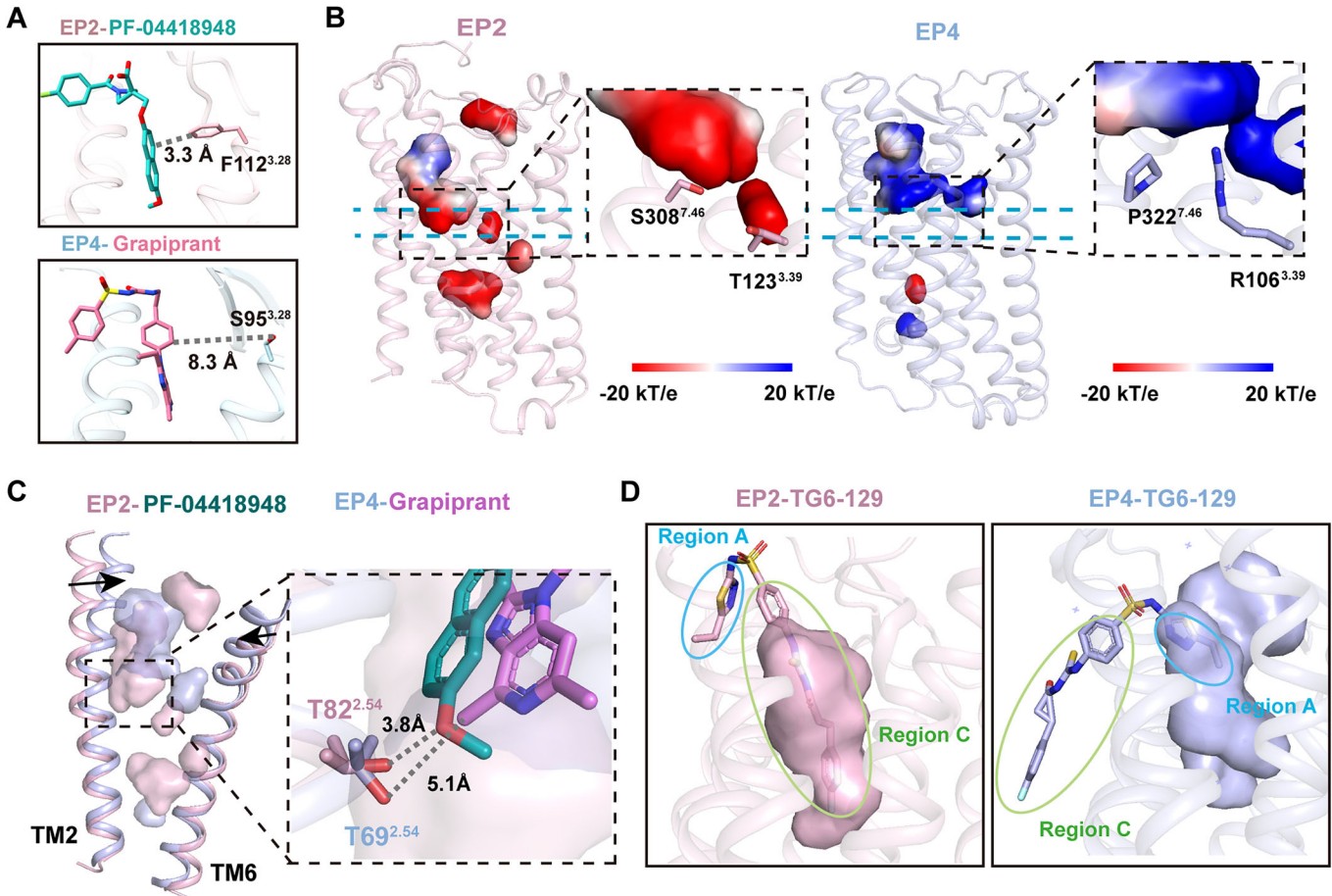

**Figure 6. Comparison of EP2 and EP4 binding pockets.**

(A) The distances between PF-04418948 and F112[3.28] in EP2 (upper panel) and between grapiprant and S95[3.28] in EP4 (bottom panel). (B) Comparison of charge characteristics between inactive EP2 and EP4 pockets, with the key residues at the bottom of the pocket, is highlighted in a zoomed-in view. (C) Relative displacement of TM2 and TM6 in EP2 and EP4. The distances between the residue position 2.54 of TM2 and PF-04418948 or grapiprant are shown in a zoomed-in view. The conformational changes are shown by black arrows. (D) The binding pocket of TG6-129 in EP2 and EP4. The Region A and Region C of TG6-129 are presented by blue and green circles, respectively.

## Implications for rational design of selective and dual EP2 and EP4 antagonists

Structural alignment of PF-04418948-bound and TG6-129-bound EP2 with active EP2 structures displayed certain distinctions. In TG6-129-bound EP2, the outward shift of TM1 and TM2, and the downward displacement of TM7 are both slightly smaller in degree (Fig. EV4A). The displacement differences are reflected in the position and conformation of some important residues in the binding pocket, including S28[1.39], Y93[3.26], T185[ECL2], and W186[ECL2] in the top of the pocket, and S86[2.58], S305[7.43], and S308[7.46] in the bottom of the pocket (Fig. EV4B). In terms of the binding pose of the two ligands, a more flexible Region B brings the sulfonyl group of TG6-129 closer to the top of the pocket, and a longer Region C of TG6-129 contributing to a more extended binding cavity than the PF-04418948-binding pocket (Fig. EV4C). The structure comparison of grapiprant-bound and TG6-129-bound EP4 revealed that the conformation of the inactive EP4 is extremely similar (Figs. 1H and EV4D). But because of the more polar and expansive

Region B, grapiprant forms close interaction with R316[7.40], Y80[3.26], and ECL2 in the top of the pocket (Fig. EV4E). Besides, the key residue S319[7.43], located inside the binding pocket and forming hydrogen bonds to both ligands, undergoes conformational changes (Fig. EV4E).

The selective binding of PF-04418948 to EP2 and grapiprant to EP4 provides crucial insights into the mechanisms underlying selective antagonism. First, the PF-04418948-bound EP2 structure reveals a unique π–π stacking interaction with the nonconserved hydrophobic residue F112[3.28] in EP2 TM3 (Fig. 6A). In contrast, EP4 has a polar residue S95[3.28] at the same position, which does not interact with grapiprant. Chimeric studies where semi-conserved regions were swapped between EP2 and EP4 were performed and revealed that these mutations affect antagonist potency but do not completely switch selectivity profiles. The I85[2.57]V and M116[3.32]L mutations in EP2 both significantly reduced the antagonistic potency of PF-04418948 and TG6-129. The L301[7.39]I mutation selectively reduced PF-04418948 antagonism while showing minimal effect on TG6-129 antagonism (Appendix Fig. S7A,B).

Conversely, only the L99$^{3.32}$M mutation in EP4 slightly decreased the antagonistic potency of grapiprant (Appendix Fig. S7C).

Second, the shape and charge characteristics of the binding pockets in EP2 and EP4 differ significantly. EP2 has a narrow and deep binding pocket with an electronegative bottom formed by polar residues T123$^{3.39}$ in TM3 and S308$^{7.46}$ in TM7, which interact with PF-04418948 or TG6-129 (Fig. 6B). In contrast, EP4 has a broader and shallower binding pocket, with an electropositive bottom where the positively charged R106$^{3.39}$ forms a barrier with the hydrophobic P322$^{7.46}$, blocking further penetration of grapiprant. The G106$^{3.39}$R mutant in EP4 retains antagonist binding but no longer binds the agonist (Toyoda et al, 2019), highlighting the importance of pocket shape and charge in selective binding.

Third, there are notable differences in the relative displacement of the TMs between EP2 and EP4. In the inactive EP4 structure, both TM2 and TM6 shift inward at the extracellular side compared to those in the inactive EP2, causing a bias in the binding pocket away from TM2 (Figs. 6C and EV3E). This shift results in T82$^{2.54}$ in EP2, maintaining an important interaction with PF-04418948, while T69$^{2.54}$ in EP4 loses interaction with grapiprant.

These differences in residue properties, pocket shape and charge, and TM displacement make the design of dual antagonists with comparable effects very challenging. However, the structure of TG6-129 bound to both receptors provides valuable structural insights. In EP2, TG6-129 is deeply inserted and tightly bound through Region C. Conversely, in EP4, TG6-129 binds via Region A and interacts only shallowly (Figs. 6D and EV3E). TG6-129 acts as dual warheads fused by a linker, with one pharmacophore binding to EP2 and the other to EP4. The binding mode of TG6-129 with EP2 and EP4 offers important ideas for designing dual antagonists.

## Discussion

Our study provides important structural insights into the selective and dual antagonism of EP2 and EP4 prostaglandin receptors, offering a valuable framework for understanding prostanoid receptor pharmacology and guiding the rational design of improved antagonists. By presenting four cryo-EM structures of human EP2 and EP4 receptors in complex with selective and dual antagonists, we have uncovered key structural features that govern antagonist selectivity and efficacy. In addition, our study also provides a structural basis for elucidating the agonistic mechanism of EP2.

Numerous studies have successfully resolved both active and inactive conformations of BRIL-fused GPCRs, including apo Frizzled5-BRIL structure (Tsutsumi et al, 2020), active serotonin receptor 5HT$_{1B}$-BRIL structures (Mukherjee et al, 2020), apo and inactive GPR61-BRIL structures (Lees et al, 2023). Its introduction in place of the intracellular loop-3 (ICL3) between TM5 and TM6 facilitates to obtain stable complex structures for cryo-EM, without any disruption to the core structure of the GPCR, demonstrating that receptor conformational states are primarily determined by bound ligands rather than the fusion construct itself.

In class A GPCRs with known ligands, position 6.48 of TM6 is predominantly occupied by tryptophan (77%), with other residues like phenylalanine (13%), glutamine/tyrosine (2.9%), serine (2%), methionine (1.5%), and alanine/valine (0.5%) occurring less frequently. Notably, serine at position 6.48 is unique to the prostaglandin receptor family. The conventional activation

mechanism in many class A GPCRs involves direct agonist interaction with W$^{6.48}$, which serves as a molecular toggle switch facilitating the characteristic outward movement of TM6 (Zhou et al, 2019). However, the W$^{6.48}$ in EP2 and EP4 are substituted with S$^{6.48}$ (Fig. 2B). There is no direct interaction between PGE$_2$ and TM6 in EP2 and EP4 (Huang et al, 2023; Qu et al, 2021). As previously reported, activation of EP2 is triggered by the ligand interactions with TM2 and TM7, which might cause a downward shift of TM7 and a series of structural rearrangements of TM7 and TM6 (Qu et al, 2021). However, there is a lack of direct comparison with inactive EP2, and the specific key residues involved in the structural rearrangements are not clear yet. Structure comparison between active PGE$_2$-EP2 and inactive PF-04418948-EP2 demonstrated a distinct propagating pathway through TM1, TM2, TM7 and TM6 involving the clash between D311$^{7.49}$ and F273$^{6.44}$ that enables the outward movement of TM6 required for EP2 activation (Fig. 2C). These findings complement the gaps in the activation mechanisms of EP2, and provide important structural basis to better understand the activation and antagonism mechanisms of EP2.

The structures of EP2 bound to PF-04418948 and EP4 bound to grapiprant reveal distinct binding pockets and interaction networks that explain their selective antagonism. The difference in residue properties at position 3.28 (F112$^{3.28}$ in EP2 and S95$^{3.28}$ in EP4), as well as the chimeric studies (Appendix Fig. S7), suggest that antagonist selectivity is determined by multiple structural features beyond individual residue differences, including broader conformational differences in the binding pockets.

Furthermore, our structures unveil significant differences in the shape and charge characteristics of the EP2 and EP4 binding pockets. EP2 possesses a narrow, deep binding pocket with an electronegative bottom, while EP4 features a broader, shallower pocket with an electropositive bottom. These distinctions in pocket architecture play a crucial role in determining antagonist selectivity and binding modes. The importance of these features is underscored by the G106$^{3.39}$R mutation in EP4, which retains antagonist binding but abolishes agonist binding (Toyoda et al, 2019), emphasizing the critical role of pocket shape and charge in ligand selectivity.

The relative displacement of TMs between EP2 and EP4 adds another layer of complexity to their binding properties. In the inactive EP4 structure, the inward shift of TM2 and TM6 at the extracellular side results in a binding pocket bias away from TM2. This structural difference explains why T82$^{2.54}$ in EP2 maintains an important interaction with PF-04418948, while the corresponding T69$^{2.54}$ in EP4 loses interaction with grapiprant. These findings highlight the dynamic property of GPCR structures and their impact on ligand binding (Zhang et al, 2024).

Our study also sheds light on the challenges of designing dual antagonists with comparable effects on both EP2 and EP4. The structure of TG6-129 bound to both receptors reveals a novel binding mode that offers promising insights for dual antagonist design. TG6-129 exhibits a deep insertion and tight binding through Region C in EP2, while in EP4, it binds shallowly via Region A. This dual warhead approach, where different parts of the molecule interact preferentially with each receptor, presents an innovative strategy for developing balanced dual antagonists.

The extensive mutagenesis studies complementing our structural data provide crucial information on the specific residues

involved in antagonist binding and selectivity. For instance, the mutations W186$^{ECL2}$A in EP2 and S319$^{7.43}$A in EP4 significantly impacted the potency of their respective antagonists, underscoring the importance of these residues in ligand recognition. These findings not only validate our structural observations but also offer valuable guidance for structure-based drug design efforts.

Our results have significant implications for drug discovery targeting EP2 and EP4 receptors. The detailed understanding of the binding modes and key interaction points of selective antagonists like PF-04418948 and grapiprant provides a solid foundation for the rational design of new, potentially more potent and selective compounds. Moreover, the unique binding mode of TG6-129 opens up new possibilities for the development of dual antagonists, which could offer enhanced therapeutic efficacy in conditions where both EP2 and EP4 signaling contribute to pathology, such as cancer and inflammatory diseases.

In conclusion, our study provides a comprehensive structural framework for understanding EP2 and EP4 antagonism, revealing the molecular basis for selective and dual inhibition. These insights not only advance our understanding of prostanoid receptor pharmacology but also pave the way for the design of next-generation antagonists with improved selectivity profiles or dual-targeting capabilities. As we continue to unravel the complexities of GPCR signaling, such structural information will be invaluable in developing more effective therapeutics targeting the PGE$_2$-EP2/EP4 signaling axis in various pathological conditions.

## Methods

### Reagents and tools table

| Reagent/resource | Reference or source | Identifier or catalog number |
| --- | --- | --- |
| **Experimental models** | | |
| *E. coli* OmniMAX 2 T1 | Invitrogen | Cat# C854003 |
| *E. coli* DH10B | Invitrogen | Cat# EC0113 |
| *Spodoptera frugiperda* (Sf9) | Expression systems | Cat#94-001 F |
| High Five | Expression systems | Cat#94-002 F |
| AD293 | ATCC | |
| **Recombinant DNA** | | |
| pFastBac EP2-BRIL | This study | N/A |
| pFastBac anti-BRIL Fab | This study | N/A |
| pET-21d anti-Fab Nanobody | This study | N/A |
| pFastBac EP4 | This study | N/A |
| pFastBac Fab001 | This study | N/A |
| pcDNA3.1 EP2 | This study | N/A |
| pcDNA3.1 EP4 | This study | N/A |

| Reagent/resource | Reference or source | Identifier or catalog number |
| --- | --- | --- |
| GloSensor plasmid | Promega | Cat# E2301 |
| **Antibodies** | | |
| Anti-FLAG FITC-conjugated antibody | Sigma | Cat# F4049 |
| **Oligonucleotides and other sequence-based reagents** | | |
| PCR primers | Tsingke | |
| **Chemicals, enzymes, and other reagents** | | |
| PF-04418948 | TargetMol | Cat# T3306 |
| Grapiprant | TargetMol | Cat# TQ0292 |
| TG6-129 | TargetMol | Cat# T28958 |
| PGE$_2$ | TargetMol | Cat# T5014 |
| ESF921 incest cell culture medium | Expression system | Cat# 96-001-20 |
| Cellfectin™ II Reagent | Gibco | Cat# 10352100 |
| Protease Inhibitor Cocktail | TargetMol | Cat# C0001 |
| Lauryl Maltose Neopentyl Glycol | Anatrace | Cat# NG310 |
| Cholesteryl Hemisuccinate Tris Salt | Anatrace | Cat# CH210 |
| Glyco-diosgenin | Anatrace | Cat# GDN101 |
| Dulbecco's Modified Eagle Medium (DMEM) | Cytiva | Cat# SH30243.01 |
| Fetal Bovine Serum | ExCell | Cat# FSP500 |
| Lipofectamine 2000 | Invitrogen | Cat# 11668019 |
| Opti-MEM | Gibco | Cat# 11058021 |
| CO2-independent medium | Gibco | Cat# 18045088 |
| **Software** | | |
| CryoSPARC | https://cryosparc.com/ | Version 4.5.1 |
| MotionCor2 | https://msg.ucsf.edu/software | Version 1.6.4 |
| PHENIX | https://phenix-online.org/ | Version 1.21.1-5286 |
| COOT | www.2.mrc-lmb.cam.ac.uk/personal/pemsley/coot/ | Version 0.9.8.93 |

| Reagent/ resource | Reference or source | Identifier or catalog number |
|---|---|---|
| UCSF Chimera | https://www.cgl.ucsf.edu/chimera/ | Version 1.17.3 |
| UCSF Chimera X | https://www.cgl.ucsf.edu/chimerax/ | Version 1.7.1 |
| PyMOL | https://pymol.org/ | Version 2.5 |
| GraphPad Prism | https://www.graphpad.com/scientific-software/prism/ | Version 8.0.1 |
| AlphaFold2 | https://colab.research.google.com/github/sokrypton/ColabFold/blob/main/AlphaFold2.ipynb#scrollTo=KK7X9T44pWb7 | ColabFold v1.5.5 |
| Other | | |
| EnVision plate reader | EnVision | |
| Guava® easyCyte flow cytometer | Guava | |

## Construction, expression, and purification of EP2-BRIL-FabBRIL-NbFab complexes

The human EP2 receptor construct was engineered to express in its inactive conformation. It incorporated an HA signal peptide, FLAG tag, and 8×His tag at the N-terminus, with BRIL inserted into intracellular loop 3 (ICL3). The BRIL insertion was positioned between ICL3 residues R224 and A257 in wild-type EP2, utilizing two brief, modified linkers derived from the A2A adenosine receptor. These linkers consisted of ARRQL (connecting R224 to the N-terminus of BRIL) and ERARSTL (linking the C-terminus of BRIL to A257). The construct was introduced into a pFastBac1 vector using the ClonExpress II One Step Cloning Kit. Additionally, anti-BRIL Fab (FabBRIL) and anti-Fab nanobody (NbFab) were engineered with an N-terminal GP67 signal peptide and a C-terminal 8×His tag, then inserted into pFastBac1 vectors.

For protein expression, High-Five insect cells were employed using the baculovirus system. Cells were cultivated in ESF 921 serum-free medium until reaching a density of 2–3 million cells per mL, at which point they were infected with EP2 baculoviruses. After incubating at 27 °C, 120 rpm for 48 h, the culture was harvested via centrifugation, and the resulting cell pellets were cryopreserved at −80 °C.

The purification process began with resuspending cell pellets in a buffer composed of 20 mM HEPES, pH 7.4, 150 mM NaCl, 10 mM $MgCl_2$, and 10 mM CaCl2, fortified with 1% (w/v) Protease Inhibitor Cocktail (TagetMol). Following homogenization by a Dounce homogenizer, membrane proteins were extracted by adding 0.5% (w/v) lauryl maltose neopentyl glycol (LMNG, Anatrace), 0.1% (w/v) cholesteryl hemisuccinate tris salt (CHS, Anatrace), and 10 µM PF-04418948 or TG6-129 antagonist for incubating 2 h at 4 °C. After removing insoluble matter by centrifugation at 30,000 rpm for 45 min, the supernatant was incubated with Ni-NTA affinity resin for 2 h at 4 °C. The resin was thoroughly washed with a wash buffer containing 20 mM HEPES, pH 7.4, 150 mM NaCl, 0.0075% (w/v) LMNG, 0.0015% CHS,

0.0025% glyco-diosgenin (GDN, Anatrace), 25 mM imidazole, 10 µM PF-04418948 or TG6-129 for 20 column volumes (CV), and then eluted with 5 CV of elution buffer (20 mM HEPES, pH 7.4, 150 mM NaCl, 0.0075% (w/v) LMNG, 0.0015% CHS, 0.0025% GDN, 300 mM imidazole, 10 µM PF-04418948 or TG6-129). The resin elution was concentrated using a 100-kDa MWCO Millipore concentrator and further refined via size exclusion chromatography on a Superdex 6 increase 10/300GL column (GE Healthcare) equilibrated with 20 mM HEPES, pH 7.4, 150 mM NaCl, 0.00075% (w/v) LMNG, 0.00015% CHS, 0.00025% (w/v) GDN, and 2 µM PF-04418948 or TG6-129.

To form the final complex, purified EP2 was combined with FabBRIL and NbFab in a 1:1.2:1.5 molar ratio and incubated on ice for 30 min. This mixture underwent a final purification step using a Superose 6 increase 10/300 GL column equilibrated with 20 mM HEPES, pH 7.4, 150 mM NaCl, 0.00075% (w/v) LMNG, 0.00015% CHS, 0.00025% (w/v) GDN, and 2 µM PF-04418948 or TG6-129. The fractions containing the EP2-FabBRIL-NbFab complex were identified by SDS-PAGE, pooled, and concentrated in preparation for cryo-EM analysis.

## Construction, expression, and purification of EP4-Fab001 complexes

The human EP4 receptor construct was engineered for expression in its inactive conformation, following previously established protocols (Toyoda et al, 2019). Key modifications included the removal of N-terminal residues 1–3, C-terminal residues 347–488, and intracellular loop 3 (residues 218–259). In addition, N-linked glycosylation sites at positions 7 and 177 were altered to glutamine. According to the previously established protocols (Toyoda et al, 2019), to achieve substantially higher stability in the inactive state, the construct incorporated two strategic point mutations: Ala62$^{2.47}$Leu and Gly106$^{3.39}$Arg. To facilitate expression and purification, an HA signal peptide was introduced at the N-terminus, while an 8×His tag was appended to the C-terminus. This optimized construct was inserted into a pFastBac1 vector using the ClonExpress II One Step Cloning Kit.

Concurrently, the anti-EP4 Fab (Fab001) was developed based on the amino acid sequence of Fab001 (Toyoda et al, 2019). The Fab001 construct was engineered to include an N-terminal GP67 signal peptide and a C-terminal 8×His tag, and subsequently cloned into a pFastBac1 vector.

The expression and purification process for human EP4 mirrored that of human EP2, with one key distinction: the ligands used were 10 µM grapiprant or TG6-129 antagonist. To form the EP4-antagonist complexes, purified EP4 was incubated with Fab001 at a molar ratio of 1:1.2 on ice for 30 min. The resulting complex underwent concentration and further purification via size exclusion chromatography, utilizing a Superose 6 increase 10/300 GL column. The chromatography buffer consisted of 20 mM HEPES, pH 7.4, 150 mM NaCl, 0.00075% (w/v) LMNG, 0.00015% CHS, 0.00025% (w/v) GDN, and 2 µM grapiprant or TG6-129.

The fractions containing the EP4-Fab001 complex were identified through SDS-PAGE analysis. These fractions were then pooled, concentrated, and prepared for subsequent cryo-EM investigations, enabling structural studies of EP4 in complex with its antagonists.

## Cryo-EM data collection

Sample preparation for cryo-EM involved the use of gold R1.2/1.3 holey carbon grids, which were glow-discharged prior to use. The Vitrobot Mark IV plunger (FEI) was employed for grid preparation, with environmental conditions set to 4 °C and 100% humidity. A 3-μL aliquot of the sample was applied to each grid, allowed to incubate for 3 s, and then blotted for 4.5 s on both sides with a blot force of 2. Immediately following blotting, the grids were rapidly plunged into liquid ethane for vitrification.

Data acquisition was performed on a Titan Krios microscope, operating at 300 kV and equipped with a Gatan K3 direct electron detector. The EPU Software (FEI Eindhoven, Netherlands) was utilized to automate the image acquisition process. For each complex—EP2-PF-04418948, EP2-TG6-129, EP4-grapiprant, and EP4-TG6-129—a substantial number of movies were collected: 5327, 6562, 5470, and 5400, respectively. The microscope was set to a magnification of 105,000, resulting in pixel sizes of 0.832 Å, 0.73 Å, 0.824 Å, and 0.824 Å for the respective datasets. Each movie comprised 36 frames, collected over a 2.5-second exposure, with a cumulative electron dose of 50 e$^-$ Å$^{-2}$.

## Cryo-EM image processing

The initial processing of cryo-EM micrographs involved drift correction using MotionCor2 (Zheng et al, 2017), followed by contrast transfer function (CTF) estimation with GCTF after importing the images into CryoSPARC v4.5.1 (Punjani et al, 2017). Subsequent analysis steps, encompassing particle picking, extraction, two-dimensional (2D) classification, Ab-Initio Reconstruction, multi-reference refinement, and local refinement, were executed using CryoSPARC v4.5.1 (Punjani et al, 2017).

For the EP2-BRIL-FabBRIL-NbFab complexes, the processing pipeline began with the extraction of 3,285,427 and 2,260,400 particles from micrographs of EP2-PF-04418948 and EP2-TG6-129 complexes, respectively. Multiple rounds of reference-free 2D classification yielded 224,164 particles for the EP2-PF-04418948 complex and 204,172 particles for the EP2-TG6-129 complex. The final structures, achieved through Ab-Initio Reconstruction, multi-reference refinement, seeding-facilitating refinement, and local refinement in CryoSPARC v4.5.1, reached resolutions of 3.50 Å for EP2-PF-04418948 and 3.28 Å for EP2-TG6-129 (Appendix Figs. S2 and S3).

The EP4-Fab001 complexes underwent a similar processing workflow. Initial particle extraction yielded 3,394,211 particles for the EP4-grapiprant complex and 3,357,544 for the EP4-TG6-129 complex. After iterative 2D classification and particle clearance, 79,765 particles remained for the EP4-grapiprant complex and 67,243 for the EP4-TG6-129 complex. The refined structures, obtained through the same series of reconstruction and refinement steps in CryoSPARC v4.5.1, achieved resolutions of 2.65 Å for EP4-grapiprant and 2.92 Å for EP4-TG6-129 (Appendix Figs. S4 and S5).

## Model building and refinement

For each of the EP2 structures and EP4 structures, an atomic model predicted by Alphafold2 was used as the starting reference model for receptor building (Tunyasuvunakool et al, 2021). Structures of FabBRIL and NbFab were derived from PDB entry 8TB7 (Lees et al,

2023), and the structure of Fab001 was derived from PDB 5YWY (Toyoda et al, 2019). They were rigid bodies that fit into the density. All models were fitted into the EM density map using UCSF Chimera (Pettersen et al, 2004), followed by iterative rounds of manual adjustment and automated rebuilding in COOT (Emsley and Cowtan, 2004) and PHENIX (Adams et al, 2004), respectively. The model was finalized by rebuilding in ISOLDE (Croll, 2018), followed by refinement in PHENIX with torsion-angle restraints to the input model. The final model statistics were validated using Comprehensive validation (cryo-EM) in PHENIX and provided in Appendix Table S1. All structural figures were prepared using Chimera, Chimera X (Pettersen et al, 2021), and PyMOL (Schrödinger, LLC.).

## GloSensor cAMP assay

The full-length EP2 or EP4 was cloned into the pcDNA3.1 vector (Invitrogen), incorporating an N-terminal FLAG tag. AD293 cells were seeded in six-well plates containing DMEM supplemented with 10% dialyzed FBS 1 day prior to transfection. Following overnight growth, cells were transfected with a 1:1 ratio of receptor construct and GloSensor22F cAMP biosensor (Promega) plasmids.

Twenty-four hours post-transfection, the cells were transferred to CO2-independent media containing 2% GloSensor cAMP Reagent (Promega) and distributed into 384-well assay plates at a density of 4000 cells per well in 10 μL. After a 1-h incubation period, 5 μL of buffer containing PGE2 and varying concentrations of test compounds was introduced to each well. The plates were then incubated at 37 °C for an additional hour before luminescence measurements were taken using an EnVision plate reader.

Data analysis was performed using GraphPad Prism software v8.0.1. A nonlinear regression analysis, employing a sigmoidal dose–response model, was conducted to determine the E$_{max}$ values and half-maximum inhibitory concentrations (IC$_{50}$) for each compound tested.

## Surface expression analysis

Cell-surface expression levels of both wild-type and mutant receptors were assessed using a fluorescence-activated cell sorting (FACS) assay. Briefly described, AD293 cells expressing FLAG-tagged EP2 or EP4 receptors were collected 24 h post-transfection. These cells were then incubated with a mouse anti-FLAG FITC-conjugated antibody (Sigma) at a 1:200 dilution for 2 h at 4 °C in a dark environment. Subsequently, cells were washed with 100 μL of PBS. Surface expression levels were quantified by measuring the fluorescence intensity of FITC using a Guava® easyCyte flow cytometer. Data were normalized relative to the expression levels of the wild-type receptors. These experiments were conducted in triplicate, and the results were expressed as mean ± S.E.M.

## Statistics

All data from functional studies were processed using GraphPad Prism v8.0.1 (Graphpad Software Inc.) and are presented as means ± S.E.M, based on a minimum of three independent experiments, each performed in triplicate. Statistical significance was assessed using a two-sided, unpaired *t* test, with a *P value of less than 0.05 considered to indicate significant differences.

## Data availability

The atomic coordinates and the cryo-EM density maps have been deposited in the Protein Data Bank (PDB) under accession numbers of 9JRO, 9JRT, 9JQZ, and 9JQY and Electron Microscopy Data Bank (EMDB) under accession codes EMD-61762, EMD-61763, EMD-61744, and EMD-61743 for the EP2-PF-04418948 complex, the EP2-TG6-129 complex, the EP4-Grapiprant complex, and the EP4-TG6-129 complex, respectively.

The source data of this paper are collected in the following database record: biostudies:S-SCDT-10_1038-S44318-025-00611-0.

## Peer review information

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

## Acknowledgements

The cryo-EM data were collected at the Shanghai Advanced Center for Electron Microscopy, Shanghai Institute of Materia Medica, Chinese Academy of Sciences. We thank QY, WH, KW, and SL for the cryo-EM data collection. This work was partially supported by The National Natural Science Foundation of China (32130022 to HEX, 82121005 to HEX, 32301016 to CW); Shanghai Post-doctoral Excellence Program (2023707 to YW); The National Key Research and Development Program of China (2022YFC2703105 to HEX); Shanghai Municipal Science and Technology Major Project (2019SHZDZX02 to HEX); Chinese Academy of Sciences Strategic Priority Research Program (XDB37030103 to HEX).

## Author contributions

**Yanli Wu**: Conceptualization; Data curation; Formal analysis; Validation; Investigation; Visualization; Methodology; Writing—original draft; Project administration; Writing—review and editing. **Heng Zhang**: Software; Formal analysis; Visualization; Methodology; Writing—original draft; Writing—review and editing. **Jiuyin Xu**: Data curation; Formal analysis; Validation; Investigation; Methodology. **Kai Wu**: Data curation. **Wen Hu**: Data curation. **Xinheng He**: Data curation; Formal analysis. **Gaoming Wang**: Data curation; Formal analysis. **Canrong Wu**: Conceptualization; Resources; Software; Supervision; Funding acquisition; Writing—original draft; Project administration; Writing—review and editing. **H Eric Xu**: Conceptualization; Resources; Supervision; Funding acquisition; Writing—original draft; Project administration; Writing—review and editing.

Source data underlying figure panels in this paper may have individual authorship assigned. Where available, figure panel/source data authorship is listed in the following database record: biostudies:S-SCDT-10_1038-S44318-025-00611-0.

## Disclosure and competing interests statement

The authors declare no competing interests.

# Expanded View Figures

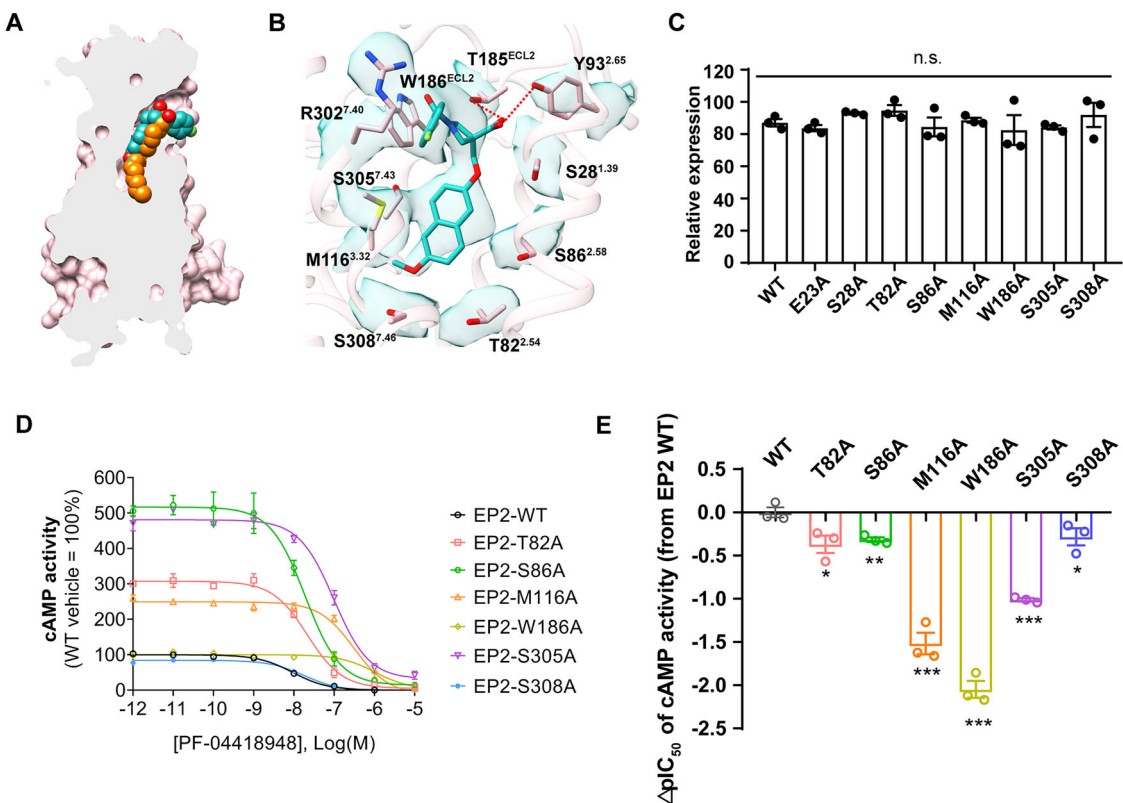

**Figure EV1.   Additional data for inhibition of EP2.**

(**A**) Comparison of binding pocket of PF-04418948 (light green) and $PGE_2$ (orange) in inactive EP2 (light pink). (**B**) Cryo-EM density map of key residues in PF-04418948 binding pocket in EP2. (**C**) Cell surface expression level of WT and mutant EP2 receptors. Data are presented as mean ± S.E.M. ($n = 3$), significance was determined with two-side unpaired $t$ test; $P > 0.05$ was considered statistically no significant (n.s.). (**D, E**) cAMP response (**D**) and $\Delta pIC_{50}$ (**E**) of PF-04418948 in constitutively active EP2 mutants. $\Delta pIC_{50} = pIC_{50}$ of PF-04418948 to specific mutant - $pIC_{50}$ of PF-04418948 to WT. Data are presented as mean ± S.E.M. of 3 independent experiments with 3 technical replicates, respectively. Significance was determined with a two-sided unpaired $t$ test. Compared with EP2 WT, the $P$ values in panel E are 0.0343, 0.0085, 0.0004, 0.0001, 0.0001 and 0.0699 for EP2 T82A, S86A, M116A, W186A, S305A and S308A mutant, respectively. *$P < 0.05$, **$P < 0.01$, ***$P < 0.001$. Source data are available online for this figure.

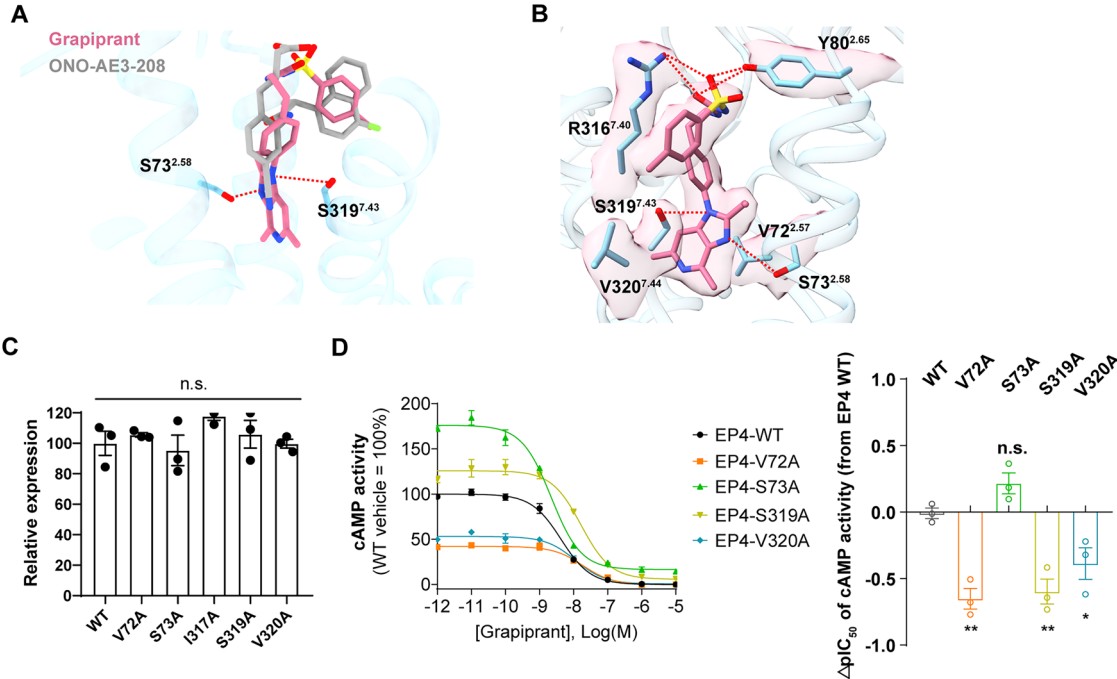

**Figure EV2.   Additional data for selective inhibition of EP4.**

(**A**) Comparison of binding pocket of grapiprant and ONO-AE3-208 in inactive EP4 (light blue). (**B**) Cryo-EM density map of key residues in grapiprant binding pocket in EP4. (**C**) Cell surface expression level of WT and mutant EP4 receptors. Data are presented as mean ± S.E.M. ($n = 3$), significance was determined with two-sided unpaired $t$ test; $P > 0.05$ was considered statistically no significant (n.s.). (**D**) cAMP response (left) and $\Delta pIC_{50}$ (right) of grapiprant in constitutively active EP4 mutants. $\Delta pIC_{50} = pIC_{50}$ of PF-04418948 to specific mutant - $pIC_{50}$ of grapiprant to WT. Data are presented as mean ± S.E.M. of 3 independent experiments with 3 technical replicates, respectively. Significance was determined with a two-sided unpaired $t$ test. Compared with EP4 WT, the $P$ values in the right panel are 0.0018, 0.0635, 0.0045 and 0.0396 for EP4 V72A, S73A, S319A and V320A mutant, respectively. *$P < 0.05$, **$P < 0.01$, ***$P < 0.001$. Source data are available online for this figure.

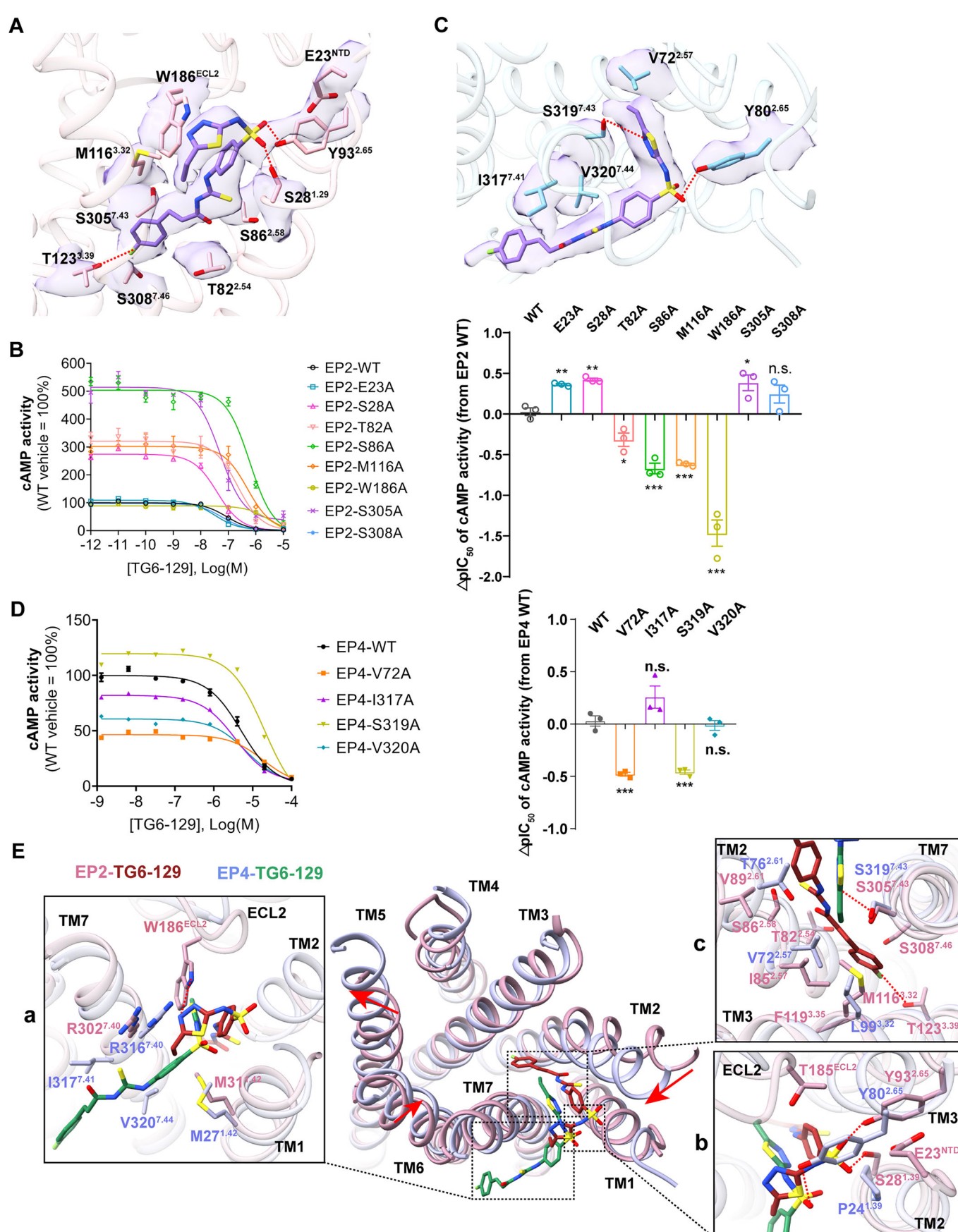

**Figure EV3. Additional data for dual inhibition of EP2 and EP4.**

(A) Cryo-EM density map of key residues in TG6-129 (purple) binding pocket in EP2 (light pink). (B) cAMP response (left) and $\Delta pIC_{50}$ (right) of TG6-129 in constitutively active EP2 mutants. $\Delta pIC_{50} = pIC_{50}$ of TG6-129 to specific mutant - $pIC_{50}$ of TG6-129 to WT. Compared with EP2 WT, the P values in the right panel are 0.0023, 0.0015, 0.0215, 0.0010, 0.0002, 0.0009, 0.0296 and 0.1455 for EP2 E23A, S28A, T82A, S86A, M116A, W186A, S305A and S308A mutant, respectively. (C) Cryo-EM density map of key residues in TG6-129 binding pocket in EP4 (light blue). (D) cAMP response (left) and $\Delta pIC_{50}$ (right) of TG6-129 in constitutively active EP4 mutants. Compared with EP4 WT, the P values in the right panel are 0.0006, 0.1223, 0.0008 and 0.5647 for EP4 V72A, I317A, S319A and V320A mutant, respectively. (E) Structure comparison of TG6-129 (brown) bound EP2 and TG6-129 (green) bound EP4. The conformational changes of top view (middle) are depicted by red arrows, and the comparison of key residues in the outside (a), top (b) and inside (c) of the binding pocket are shown in a zoomed-in view. Data information: Data are presented as mean ± S.E.M. of 3 independent experiments with 3 technical replicates respectively. Significance was determined with a two-sided unpaired *t* test; *$P \leq 0.05$, **$P \leq 0.01$, ***$P \leq 0.001$, n.s. $P > 0.05$. Source data are available online for this figure.

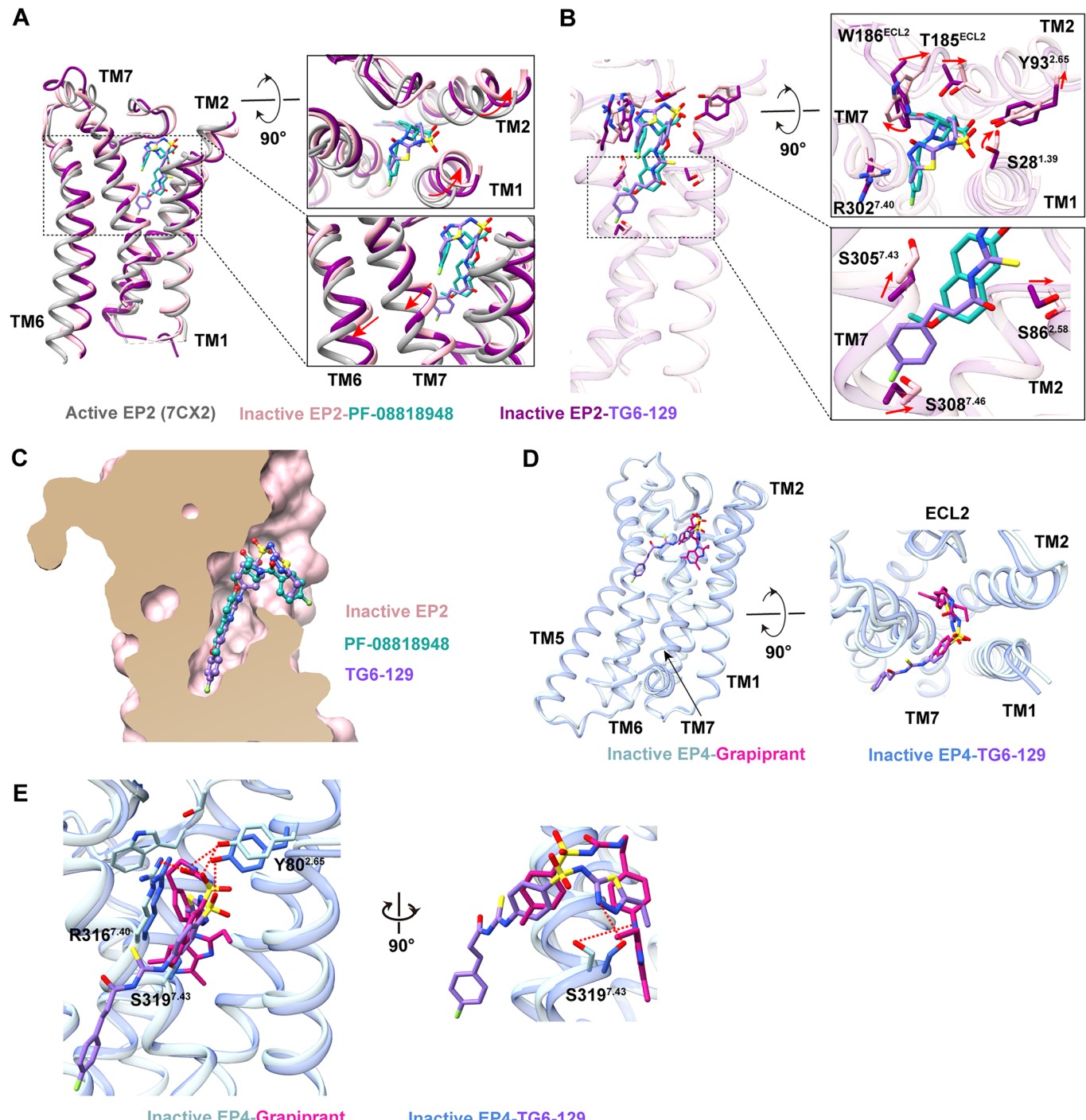

**Figure EV4. Additional data for rational design of selective and dual EP2 and EP4 antagonists.**

(A) Structure comparison of active EP2, inactive PF-04418948-EP2 and inactive TG6-129-EP2 complexes. The conformational changes of top view (upper right) and in the TM6, TM7 (bottom right) are depicted by red arrows in a zoomed-in view. (B) Comparison of the key residues in EP2 interacting with PF-04418948 and TG6-129. The residue position and conformational changes of in the top of the pocket (upper right) and in the TM6, TM7 (bottom right) are depicted by red arrows in a zoomed-in view. (C) Comparison of binding pocket of PF-04418948 and TG6-129 in inactive EP2. (D) Side view (left) and top view (right) of structure comparison between inactive grapiprant-EP4 and inactive TG6-129-EP4 structures. (E) Comparison of the key residues in EP2 interacting with PF-04418948 and TG6-129 in the front (left) and side (right) view.

