## [Peer Review File · The EMBO Journal]

Structural Insights into Selective and Dual Antagonism of EP2 and EP4 Prostaglandin Receptors

Yanli Wu, Heng Zhang, Jiuyin Xu, Kai Wu, Wen Hu, Xinheng He, Gaoming Wang, Canrong Wu, and H. Eric Xu

Corresponding authors: H. Eric Xu (eric.xu@simm.ac.cn), Canrong Wu (wcr13215@rjh.com.cn), Yanli Wu (wuyanli@simm.ac.cn)

Review Timeline:

Submission Date:	11th Jun 25
Editorial Decision:	18th Jul 25
Revision Received:	15th Aug 25
Editorial Decision:	10th Sep 25
Revision Received:	23rd Sep 25
Accepted:	2nd Oct 25

Editor: Ieva Gailite

Transaction Report:

Dear Dr. Xu,

We have now received comments from a full set of reviewers, which are included below for your information.

As you will see, all reviewers are generally positive in their evaluation and appreciate the contribution of the study to the research field. However, they also ask for further validation of the cAMP inhibition assays (reviewer #1), better description and comparison of the cryo-EM structures (reviewers #1 and #3) and a validation of the distinct TG6-129 binding mode to EP2 and EP4 with mutational analysis (reviewer #3).

From my side, I find these requests generally reasonable. Therefore, based on these positive assessments, I invite you to submit a revised manuscript in response to the comments by all reviewers. I think that it would be useful to discuss the revision in more detail via email or phone/videoconferencing - please let me know which option you prefer.

We generally allow three months as standard revision time, which can be extended to six months in the case of major revisions. Should you foresee a problem in meeting this deadline, please let us know in advance to discuss an extension.

As a matter of policy, competing manuscripts published during this period will not negatively impact on our assessment of the conceptual advance presented by your study. However, please contact me as soon as possible upon publication of any related work to discuss the appropriate course of action.

When preparing your letter of response to the referees' comments, please bear in mind that this will form part of the Review Process File and will therefore be available online to the community. For more details on our Transparent Editorial Process, please visit our website: <https://www.embopress.org/page/journal/14602075/authorguide#transparentprocess>. Please also see the attached instructions for further guidelines on preparation of the revised manuscript.

Please feel free to contact me if you have any further questions regarding the revision. Thank you for the opportunity to consider your work for publication. I look forward to discussing your revision.

With best regards,

Ieva Gailite

- a point-by-point response to the referees' comments, with a detailed description of the changes made (as a word file).
- a word file of the manuscript text.
- individual production quality figure files (one file per figure)
- a complete author checklist, which you can download from our author guidelines (<https://www.embopress.org/page/journal/14602075/authorguide>).

- Expanded View files (replacing Supplementary Information)

We realize that it is difficult to revise to a specific deadline. In the interest of protecting the conceptual advance provided by the work, we recommend a revision within 3 months (16th Oct 2025). Please discuss the revision progress ahead of this time with the editor if you require more time to complete the revisions.

Referee #1:

This manuscript reports high-resolution cryo-EM structures of the human PGE₂ receptors EP2 and EP4 in complex with selective antagonists PF-04418948 and grapiprant, as well as the dual antagonist TG6-129. The study provides valuable structural insights into the ligand binding modes and selectivity determinants of these receptors. PF-04418948 binds deeply in the EP2 pocket, stabilizing an inactive conformation, whereas grapiprant engages EP4 more superficially, forming interactions with EP4-specific residues. TG6-129 exhibits a dual binding mode that underlies its differential potency. Structural comparisons between active and inactive EP2 receptors reveal a unique activation mechanism distinct from canonical class A GPCRs, supported by mutagenesis and functional data. Overall, the work establishes a solid structural framework for rational design of selective and dual EP2/EP4 antagonists, with therapeutic implications in inflammation and cancer.

Comments:

1. A major concern is regarding the cAMP inhibition assays presented in Figures 3d, 4e, and 5d, and related supplementary figures. According to the methods, a fixed concentration of PGE₂ was used to stimulate cAMP production, and the inhibition potency of each antagonist was assessed for various receptor mutants. However, mutations may also affect the potency and efficacy of PGE₂ itself (highly likely). Without assessing PGE₂-induced responses for each mutant, the derived IC₅₀ values of antagonists may not be directly comparable. For instance, a mutation that weakens PGE₂ potency could lead to an apparent change in antagonist activity, even if antagonist binding is unaffected by such mutation. It is essential to include potency and efficacy data for PGE₂ on each mutant to validate these comparisons. Analysis of the effect of each mutation on the binding of antagonists needs to include such PGE₂ data as well.

2. Although the selectivity profiles of PF-04418948, grapiprant, and TG6-129 have been previously reported, it is important to include functional validation using the cAMP assay system employed in this study. Given that a major focus of the work is on elucidating structural determinants of ligand selectivity, providing direct experimental evidence that PF-04418948 and grapiprant are subtype-selective while TG6-129 exhibits dual activity would be important. Additionally, it would be useful to clarify whether TG6-129 displays comparable potency at both EP2 and EP4, or retains some degree of selectivity.

3. Cryo-EM maps in Fig. 1a and 1d are potentially misleading, as they imply simultaneous binding of two different ligands to the same receptor. It would be more appropriate to show separate cryo-EM maps for each ligand-receptor complex. A single cryo-EM map cannot represent both (quote) "EP2 complexes with PF-04418948 or TG6-129."

4. The EP2 structures were determined using a BRIL insertion into ICL3, which may constrain the movement of TM5 and TM6. Although this strategy is common in structural studies of inactive GPCRs, additional evidence such as ligand binding data or MD simulations would strengthen the argument that the observed conformation reflects or closely resembles a native inactive state. This is in particular important since the authors used the structures in their structural comparison analysis to dissect receptor activation mechanisms.

5. A structural overlay of the two ligands bound to EP2 or EP4 would be helpful to better show the differences in binding poses and interaction networks. This addition would enhance understanding of how these ligands achieve their selectivity by adopting different binding modes.

6. Supplementary Figures 5b and 7 show cAMP dose-response curves, but only the Δ -pIC50 values are reported in the main text. Full dose-response curves contain much more information, such as Hill slope, curve fitting quality and confidence, and whether full or partial inhibition is achieved, which are equally important for interpreting GPCR pharmacology. Reporting only Δ -pIC50 values oversimplifies the data and omits key information. This practice, while seen in some GPCR structural biology papers, should be reconsidered in favor of a more rigorous pharmacological presentation.

7. Cryo-EM density maps for key residues within the ligand-binding pockets, particularly those probed by mutagenesis, should be shown.

Referee #2:

This manuscript presents cryo-electron microscopy (cryo-EM) structures of human EP2 and EP4 prostaglandin E2 (PGE2) receptors bound to selective antagonists PF-04418948 (for EP2) and grapiprant (for EP4), as well as the dual antagonist TG6-129. The study elucidates the molecular mechanisms underlying selective and dual antagonism, revealing distinct binding pockets and interaction networks that determine antagonist specificity and efficacy. By comparing antagonist-bound (inactive) structures with PGE2-bound (active) states, the authors provide insights into the conformational changes associated with receptor activation. Extensive mutagenesis studies validate the structural findings, confirming the roles of key residues in antagonist binding and selectivity. The resolutions of the cryo-EM structures range from 2.65 Å to 3.50 Å, enabling precise modeling of ligand-receptor interactions.

This study represents a significant advancement in the fields of G protein-coupled receptor (GPCR) pharmacology and structural biology. EP2 and EP4 receptors are critical therapeutic targets due to their involvement in inflammation, pain, and cancer progression (Prostaglandin EP2 receptor: Novel therapeutic target). By providing high-resolution structures of these receptors in complex with both selective and dual antagonists, the authors offer a comprehensive framework for understanding how these receptors can be modulated pharmacologically. This is particularly relevant for developing therapies with improved selectivity and efficacy for conditions such as inflammatory diseases and cancer.

I recommend acceptance of this manuscript after minor revisions to address the points outlined below. The study makes a substantial contribution to the understanding of EP2 and EP4 receptor pharmacology and provides a structural blueprint for developing next-generation therapeutics. It will be of broad interest to researchers in structural biology, pharmacology, and drug design.

While the manuscript is generally well-written, a few minor revisions could enhance its clarity and accessibility:

Main Text Clarity: In the Results section, phrases like "The antagonist PF-04418948 binds to EP2 with high affinity" could be more precise by including the specific affinity value (e.g., Kd) or referencing the relevant figure or table. This would help readers quickly grasp the quantitative aspects of the findings.

Figure Legends: The legend for Figure 2, which likely depicts cryo-EM density maps, could be expanded to include brief descriptions of each panel. For example, clarifying whether a panel shows the overall structure or a zoomed-in view of the binding pocket would aid readers unfamiliar with cryo-EM data.

Methods Section: The authors describe the use of software for structure determination but omit the versions of these tools. Including software versions (e.g., RELION, CryoSPARC) would improve reproducibility, as updates can affect outcomes.

Typographical Errors: Minor typos, such as "receptros" instead of "receptors" on page 5, should be corrected to maintain professionalism.

Supplementary Data: A supplementary table summarizing all mutants tested and their effects on antagonist binding would be a valuable addition. This would serve as a comprehensive reference for readers interested in the mutagenesis data.

Referee #3:

Wu et al have determined four new cryo-EM structures of the EP2 and EP4 prostaglandin receptors, all in the inactive antagonist-bound state. The structures of EP2 are bound to either PF04418948 or TG6-129, and the EP4 receptor structures are bound to either grapiprant or TG6-129. Previously published structures of the EP4 receptor included two structures in the inactive state (Toyoda et al 2018) and six structures in an active state coupled to G proteins and bound to agonists (Nojima et al 2020, Huang et al 2024). By contrast, there were only three previously published structures of the EP2 receptor, all in the G protein coupled state and bound to agonist (Qu et al 2021). There are thus two aspects of the current manuscript that make it interesting. Firstly, the inactive state of the EP2 receptor, which allows a refinement of the activation mechanism (although this is essentially identical to the EP4 receptor previously published). Secondly, and more interestingly, the structures of the EP2 and EP4 receptors in the inactive state and bound to the identical antagonist, TG6-129. This represents a particularly striking example of a 'two-warhead' ligand, where one end binds and inhibits EP2, whilst the other end binds and inhibits EP4. This

property of TG6-129 appears to be totally serendipitous, as the original paper describing its synthesis and characterisation was aiming only for an EP2-specific antagonist (Ganesh et al 2013).

The manuscript, on the whole, is reasonably clear and well written, amply supported by the figures. The first part of the results section focusses on describing the selective antagonist structures, with the final half describing the dual inhibitor structures. As this section is the crux of the whole paper, it needs some more work to make the comparisons between the structures clearer (see below).

Major points

1. Figure 5 shows the structures of EP2 bound to TG6-129 and EP4 bound to TG6-129, but there are no panels that show direct comparisons between the two receptors bound to TG6-129 or a comparison of which residues bind the ligand. A key panel to add is an overlay of the two receptors and then highlighting (in sticks) the different orientations and positions of TG6-129 in the different receptors. Another comparison that needs to be made is to have one panel that shows side-by-side the amino acid residues interacting with TG6-129 and which helices they are in.
2. It would also be helpful to show overlays of other combinations of ligands after alignment of receptors e.g. TG6-129 and grapiprant and PF-04418948.
3. The apparent resolutions quoted for the cryo-EM structures need to be for just the receptor portion of the complex as well as for the whole complex. This is important so that the reader gets an accurate description of the resolution of the receptor that is not over-inflated by the well-ordered regions of the bound antibody (which sometimes happens). Two figures therefore need to be quoted in Table S1 (map resolution of the complex and map resolution of the receptor. In addition, the local resolution figures (Sup Figs 2 and 3) need to be scaled appropriately so that the density shows a range of colour and not just all blue, which is highly uninformative. There needs to be roughly equal amounts of dark blue and dark red in each figure.
4. The different binding mode of TG6-129 to EP2 and EP4 is the key result in this manuscript, yet despite the claim in the manuscript of 'extensive' mutagenesis studies, this binding mode has not been validated according to the authors own criteria (as depicted in Figures 3d and 4e). For consistency and completeness, the authors should mutate all the residues in the binding pocket of EP4 and EP2 (as defined in Figs 5c and 5i) to alanine and test binding of TG6-129. It will be important to include residues in the base of the pocket of EP4 to show that the ligand does not access deeper regions where other antagonists can bind.
5. The methods section for the purification of the receptors is too brief and incomplete for any competent researcher to follow and repeat, so this needs to be expanded for both receptors to include full buffer compositions, temperatures, times of centrifugation etc. For example, it is unacceptable to write 'Throughout this process, a specialized buffer containing LMNG, GDN, digitonin, and CHS was utilized to maintain protein stability and ligand binding'. It is a requirement for publication that full methods are given.

Minor points

No page numbers or line numbers are given, so I will not correct any of the many typographical errors or areas of ambiguity in phraseology as it is too difficult to do.

1. Standard nomenclature for G proteins requires the subscripting of the q, s and i descriptors. Please do this throughout the text.
2. For the first mention of Fab001 in the results section, please quote the reference (number 24)
3. Emory University should not be quoted for the development of TG6-129, cite the authors involved
4. In the methods, state how the mutations A62L and G106R were chosen and what is their purpose
5. All the figures are too small. The minimum text size should be 5 font (non-serif). In many instances it is impossible to read the text because it is so small and pixellated (e.g. the FSC curves in Fig S2 and S3).
6. In all the legends to structure figures, colours are cited according to their names in Chimera/ChimeraX and are often very unhelpful (what colour is peru?). Please use standard colour names e.g blue, green, turquoise and define as either light or dark.
7. For all graphs showing the activity of mutants, make a separate Table in supplementary giving the average values plotted, the errors, the number of replicates, the actual P values, and (where appropriate) the IC50s, EC50s and Emax values. The number of biological replicates are cited, but how many technical replicates were included in each measurement; please state in every case.
8. Fig 1e legend: The colours associated with the hydrophobic and polar charged residues are mislabelled. Histidine should be included as a polar charged residue (pka ~6.5, buffers used pH 7.4). so about ~90% is charged.
9. Wild type data needs to be included in the bar graphs in Fig 3d and 4e to show data points and errors.

Manuscript ID: EMBOJ-2025-121599

Title: Structural Insights into Selective and Dual Antagonism of EP2 and EP4 Prostaglandin Receptors

We thank the referees for their positive assessments on the quality and importance of our work. Their constructive suggestions have been invaluable in revising our manuscript.

In the following sections, we provide point-by-point responses to the comments made by the three referees of our original paper. The referee's comments are presented in **black** and our responses are in **blue**.

Point-by-point responses to Referee #1

This manuscript reports high-resolution cryo-EM structures of the human PGE₂ receptors EP2 and EP4 in complex with selective antagonists PF-04418948 and grapiprant, as well as the dual antagonist TG6-129. The study provides valuable structural insights into the ligand binding modes and selectivity determinants of these receptors. PF-04418948 binds deeply in the EP2 pocket, stabilizing an inactive conformation, whereas grapiprant engages EP4 more superficially, forming interactions with EP4-specific residues. TG6-129 exhibits a dual binding mode that underlies its differential potency. Structural comparisons between active and inactive EP2 receptors reveal a unique activation mechanism distinct from canonical class A GPCRs, supported by mutagenesis and functional data. Overall, the work establishes a solid structural framework for rational design of selective and dual EP2/EP4 antagonists, with therapeutic implications in inflammation and cancer.

Comments:

1. A major concern is regarding the cAMP inhibition assays presented in Figures 3d, 4e, and 5d, and related supplementary figures. According to the methods, a fixed concentration of PGE₂ was used to stimulate cAMP production, and the inhibition potency of each antagonist was assessed for various receptor mutants. However, mutations may also affect the potency and efficacy of PGE₂ itself (highly likely). Without assessing PGE₂-induced responses for each mutant, the derived IC₅₀ values of antagonists may not be directly comparable. For instance, a mutation that weakens PGE₂ potency could lead to an apparent change in antagonist activity, even if antagonist binding is unaffected by such mutation. It is essential to include potency and efficacy data for PGE₂ on each mutant to validate these comparisons. Analysis of the effect of each mutation on the binding of antagonists needs to include such PGE₂ data as well.

Response: We thank the referee for this insightful suggestion regarding the potential confounding effects of mutations on PGE₂-induced responses. We agree that mutations of key residues may affect both antagonist binding and PGE₂ potency/efficacy, which could complicate the interpretation of IC₅₀ values when using PGE₂-stimulated assays. As the referee correctly points out, and as previously reported, mutations such as T82A, S86A, M116A, and S305A in EP2 indeed weaken the binding affinity and potency of PGE₂ (Qu *et al* 2021).

To address this important methodological concern and eliminate the potential confounding effects of altered PGE₂ responses, we conducted additional experiments using constitutively active EP2 and EP4 receptors. Since both EP2 and EP4 exhibit constitutive activity, this approach allows us to assess antagonist potency independently of agonist stimulation, thereby avoiding the complication of

mutation-dependent changes in PGE₂ responses. While direct binding assays would be ideal, the technical challenges associated with developing reliable radioligand binding assays for these antagonists precluded this approach. The constitutive activity inhibition assays, presented in **Figures EV1C, D, EV2D, and EV3B, D**, provide a robust functional readout that circumvents the PGE₂-dependent confounding effects.

Importantly, our comparative analysis demonstrated that most alanine mutations showed consistent effects on antagonist potency across both experimental paradigms (PGE₂-stimulated and constitutively active receptor assays). This concordance strongly validates our conclusions regarding the critical roles of these key residues in antagonist binding and provides confidence that the observed effects are primarily due to direct impacts on antagonist-receptor interactions rather than indirect effects through altered PGE₂ responses.

2. Although the selectivity profiles of PF-04418948, grapiprant, and TG6-129 have been previously reported, it is important to include functional validation using the cAMP assay system employed in this study. Given that a major focus of the work is on elucidating structural determinants of ligand selectivity, providing direct experimental evidence that PF-04418948 and grapiprant are subtype-selective while TG6-129 exhibits dual activity would be important. Additionally, it would be useful to clarify whether TG6-129 displays comparable potency at both EP2 and EP4, or retains some degree of selectivity.

Response: We thank the referee for bringing up this important point. Given that ligand selectivity is a central focus of our study, we agree that it is essential to provide direct experimental validation of the selectivity profiles using our own assay system rather than relying solely on previously published data.

To address this concern, we conducted comprehensive selectivity profiling experiments to evaluate the antagonistic activity of all three ligands (PF-04418948, grapiprant, and TG6-129) against both EP2 and EP4 receptors using the GloSensor cAMP assay employed throughout our study. These selectivity data are now presented in **Appendix Figure S1A** and provide clear experimental evidence that: (1) PF-04418948 is highly selective for EP2 over EP4; (2) grapiprant demonstrates strong selectivity for EP4 over EP2; (3) TG6-129 exhibits dual antagonistic activity against both receptors but with preferential potency toward EP2.

This direct functional validation using our experimental system not only confirms the previously reported selectivity profiles but also provides quantitative data on the degree of selectivity that directly supports our structural interpretations. The differential potency of TG6-129 at EP2 versus EP4 is particularly relevant to understanding its unique dual-warhead binding mechanism revealed by our structural studies.

3. Cryo-EM maps in Fig. 1a and 1d are potentially misleading, as they imply simultaneous binding of two different ligands to the same receptor. It would be more appropriate to show separate cryo-EM maps for each ligand-receptor complex. A single cryo-EM map cannot represent both (quote) "EP2 complexes with PF-04418948 or TG6-129."

Response: We thank the referee for this valuable suggestion regarding the proper presentation of cryo-EM density maps. We agree that the original figure layout was potentially misleading and could

incorrectly suggest simultaneous binding of multiple ligands to a single receptor complex.

To address this concern and ensure clarity, we have restructured Figure 1 to show separate cryo-EM density maps for each individual ligand-receptor complex. The revised figure now presents: the PF-04418948-EP2 complex in **Fig. 1A**, the TG6-129-EP2 complex in **Fig. 1B**, the grapiprant-EP4 complex in **Fig. 1F (left)**, and the TG6-129-EP4 complex in **Fig. 1F (right)**, eliminating any potential ambiguity about the stoichiometry or binding arrangements.

This revised presentation provides a more accurate and scientifically rigorous representation of our structural data, with each cryo-EM map unambiguously corresponding to one specific ligand-receptor complex.

4. The EP2 structures were determined using a BRIL insertion into ICL3, which may constrain the movement of TM5 and TM6. Although this strategy is common in structural studies of inactive GPCRs, additional evidence such as ligand binding data or MD simulations would strengthen the argument that the observed conformation reflects or closely resembles a native inactive state. This is in particular important since the authors used the structures in their structural comparison analysis to dissect receptor activation mechanisms.

Response: We appreciate the referee's insightful comment regarding the potential conformational constraint imposed by the BRIL insertion, especially on the movement of TM5 and TM6, and the importance of verifying that the solved structure indeed reflects an inactive-like state.

To address this concern, we performed molecular dynamics (MD) simulations starting from the BRIL-inserted EP2 structure. Rather than relying on global RMSD, we focused our analysis on a well-established structural hallmark of GPCR activation: the intracellular distance between residues R^{3.50} and L^{6.34}, which corresponds to the relative positioning of TM3 and TM6.

Throughout the course of the simulation (500 ns × 3 repetitions), this distance remained consistently close to 8.0 Å, which is characteristic of the inactive state and significantly smaller than the typical values observed in active GPCR structures (usually >13–14 Å). This stability suggests that no substantial outward movement of TM6 occurred, and that the receptor maintained an inactive-like conformation.

This MD simulations result is now displayed in **Appendix Figure S6A**, providing strong evidence that the BRIL-inserted EP2 structure used in our structural comparison reflects a native-like inactive state (see **page 8, line 167-170**).

5. A structural overlay of the two ligands bound to EP2 or EP4 would be helpful to better show the differences in binding poses and interaction networks. This addition would enhance understanding of how these ligands achieve their selectivity by adopting different binding modes.

Response: We appreciate the referee's insightful suggestion to include structural overlays that would facilitate direct comparison of ligand binding modes and interaction networks. Such comparisons are indeed valuable for understanding the molecular basis of ligand selectivity.

To address this request, we have performed comprehensive structural alignments and overlays to highlight the differences in binding poses and interaction patterns. Specifically, we conducted structural alignments of PF-04418948-bound and TG6-129-bound EP2 structures with the active EP2 structure, which revealed distinct differences in transmembrane helix displacement, key residue positioning and conformations, and ligand superposition within the binding pocket. These comparative analyses are presented in **Figure EV4A-C**.

Additionally, we performed structural alignment of the grapiprant-bound and TG6-129-bound EP4 structures, which demonstrated remarkably similar overall conformations despite the different ligands, as shown in **Figure EV4D, E**. This comparison provides important insights into how EP4 can accommodate different antagonists while maintaining a consistent inactive conformation.

These structural overlays clearly illustrate how the different ligands achieve their selectivity through distinct binding modes and interaction networks, with each ligand optimally positioned to engage receptor-specific residues and pocket geometries. The comparative analysis supports our conclusions about the molecular determinants of antagonist selectivity and is described in detail in the main text (see **new paragraph on page 14-15, lines 358-373**).

6. Supplementary Figures 5b and 7 show cAMP dose-response curves, but only the Δ -pIC₅₀ values are reported in the main text. Full dose-response curves contain much more information, such as Hill slope, curve fitting quality and confidence, and whether full or partial inhibition is achieved, which are equally important for interpreting GPCR pharmacology. Reporting only Δ -pIC₅₀ values oversimplifies the data and omits key information. This practice, while seen in some GPCR structural biology papers, should be reconsidered in favor of a more rigorous pharmacological presentation.

Response: We appreciate the referee's insightful suggestion regarding the importance of comprehensive pharmacological data presentation. We completely agree that full dose-response curves provide critical pharmacological information beyond simple potency values, including Hill slopes, curve fitting quality, efficacy parameters, and the extent of inhibition achieved—all of which are essential for proper interpretation of GPCR pharmacology and mechanism of action.

To address this concern and provide a more rigorous pharmacological presentation, we have made key changes to our data presentation:

1. **Enhanced main text figures:** We have moved the complete dose-response curves from the supplementary material to the main text figures (**Fig. 3D, 4E, 5D, and 5K**), allowing readers to directly assess curve quality and the extent of inhibition for each mutant.
2. **Comprehensive pharmacological parameters:** We have compiled detailed pharmacological data including precise pIC₅₀ values, E_{max} values and statistical significance values (actual p values) in **Appendix Table S3, S4**. This provides the complete pharmacological profile for each compound-receptor interaction.

We believe this more comprehensive presentation significantly strengthens the pharmacological rigor of our study and provides readers with the complete dataset necessary for proper interpretation of our structure-activity relationships.

7. Cryo-EM density maps for key residues within the ligand-binding pockets, particularly those probed

by mutagenesis, should be shown.

Response: We appreciate the referee's insightful suggestion regarding the importance of showing experimental density for key residues in the ligand-binding pockets. Providing clear visualization of the cryo-EM density for structurally and functionally important residues is essential for validating our structural interpretations and supporting the mutagenesis data.

To address this request, we have prepared detailed density maps specifically highlighting key residues within the binding pockets that are critical for ligand recognition and binding. These include residues that form important hydrogen bonds and other specific interactions with the ligands, as well as residues that were systematically probed through our mutagenesis studies. The density quality for these critical residues demonstrates the reliability of our structural models and supports the structure-function relationships we have identified.

These detailed density visualizations are now presented in **Figure EV1B, EV2B, EV3A, C**, with each figure showing clear, well-resolved density for the key interaction residues in the respective ligand-receptor complexes. The quality of the density maps for these functionally important residues provides strong experimental support for our structural interpretations and validates the molecular interactions that underlie our mutagenesis results.

Point-by-point responses to Referee #2

This manuscript presents cryo-electron microscopy (cryo-EM) structures of human EP2 and EP4 prostaglandin E2 (PGE₂) receptors bound to selective antagonists PF-04418948 (for EP2) and grapiprant (for EP4), as well as the dual antagonist TG6-129. The study elucidates the molecular mechanisms underlying selective and dual antagonism, revealing distinct binding pockets and interaction networks that determine antagonist specificity and efficacy. By comparing antagonist-bound (inactive) structures with PGE₂-bound (active) states, the authors provide insights into the conformational changes associated with receptor activation. Extensive mutagenesis studies validate the structural findings, confirming the roles of key residues in antagonist binding and selectivity. The resolutions of the cryo-EM structures range from 2.65 Å to 3.50 Å, enabling precise modeling of ligand-receptor interactions.

This study represents a significant advancement in the fields of G protein-coupled receptor (GPCR) pharmacology and structural biology. EP2 and EP4 receptors are critical therapeutic targets due to their involvement in inflammation, pain, and cancer progression (Prostaglandin EP2 receptor: Novel therapeutic target). By providing high-resolution structures of these receptors in complex with both selective and dual antagonists, the authors offer a comprehensive framework for understanding how these receptors can be modulated pharmacologically. This is particularly relevant for developing therapies with improved selectivity and efficacy for conditions such as inflammatory diseases and cancer.

I recommend acceptance of this manuscript after minor revisions to address the points outlined below. The study makes a substantial contribution to the understanding of EP2 and EP4 receptor pharmacology and provides a structural blueprint for developing next-generation therapeutics. It will be of broad interest to researchers in structural biology, pharmacology, and drug design.

While the manuscript is generally well-written, a few minor revisions could enhance its clarity and accessibility:

Main Text Clarity: In the Results section, phrases like "The antagonist PF-04418948 binds to EP2 with high affinity" could be more precise by including the specific affinity value (e.g., Kd) or referencing the relevant figure or table. This would help readers quickly grasp the quantitative aspects of the findings.

Response: We appreciate the referee's useful suggestion regarding the need for more precise quantitative information in the main text. While direct binding affinity (Kd) values are not available for these antagonists, we have incorporated the most relevant quantitative potency data from the literature and our own functional studies to provide readers with immediate quantitative context.

Specifically, the reported KB value (functional potency) of PF-04418948 against PGE₂-induced cAMP signaling in EP2 is 1.8 nM using the GloSensor cAMP assay. This quantitative information has been added to the Results section (see text on page 8, line 184-187). Additionally, we have included the functional potency value of grapiprant against EP4 (K_i = 13 nM) in the main text (page 10, lines 249-251) to provide comparable quantitative parameters for both selective antagonists.

Figure Legends: The legend for Figure 2, which likely depicts cryo-EM density maps, could be expanded to include brief descriptions of each panel. For example, clarifying whether a panel shows the overall structure or a zoomed-in view of the binding pocket would aid readers unfamiliar with cryo-EM data.

Response: We appreciate the referee's helpful suggestion regarding the clarity of figure legends for readers who may be less familiar with cryo-EM data interpretation. To address this concern, we have significantly expanded the legend for Figure 1, which presents the cryo-EM density maps of our ligand-receptor complexes.

The revised legend now includes detailed descriptions of each panel, clearly specifying whether each shows the overall complex structure, zoomed-in views of the binding pocket, or specific structural features. We have also clarified the color coding scheme and provided brief explanations of the structural elements to guide readers through the cryo-EM data interpretation. These enhancements will aid readers in understanding the structural information presented and improve the overall accessibility of our structural data.

Methods Section: The authors describe the use of software for structure determination but omit the versions of these tools. Including software versions (e.g., RELION, CryoSPARC) would improve reproducibility, as updates can affect outcomes.

Response: We appreciate the referee's useful suggestion regarding software version reporting, which is indeed critical for reproducibility in computational structural biology. Software updates can significantly affect data processing workflows and final outcomes, making version specification essential for other researchers to replicate our work.

We have added specific software versions for all programs used in structure determination, including

CryoSPARC v4.5.1, MotionCor2, UCSF Chimera, ChimeraX, COOT, PHENIX, and ISOLDE in the Methods section (see **Regent Tools Table**). This information will enable other researchers to reproduce our data processing and structure refinement protocols accurately.

Typographical Errors: Minor typos, such as "receptros" instead of "receptors" on page 5, should be corrected to maintain professionalism.

Response: We thank the referee for the careful review and for identifying this typographical error. We have corrected the misspelling of "receptors" on page 5 and have conducted a comprehensive proofreading of the entire manuscript to identify and correct any additional typographical errors or inconsistencies. We have implemented a systematic review process to ensure the professional presentation of our work.

Supplementary Data: A supplementary table summarizing all mutants tested and their effects on antagonist binding would be a valuable addition. This would serve as a comprehensive reference for readers interested in the mutagenesis data.

Response: We thank the referee for this valuable suggestion. A comprehensive supplementary table would indeed serve as an important reference for the mutagenesis community and researchers interested in structure-activity relationships.

We have created **Appendix Table S2-S5**, which provide complete summaries of all mutants tested and their effects on antagonist binding. This table includes detailed quantitative data such as pIC₅₀ values, E_{max} values, statistical significance (actual p-values), and the magnitude of effect for each mutation. This comprehensive reference will facilitate comparison of mutational effects across different antagonists and enable researchers to identify key residues for future studies.

Point-by-point responses to Referee #3

Wu et al have determined four new cryo-EM structures of the EP2 and EP4 prostaglandin receptors, all in the inactive antagonist-bound state. The structures of EP2 are bound to either PF04418948 or TG6-129, and the EP4 receptor structures are bound to either grapiprant or TG6-129. Previously published structures of the EP4 receptor included two structures in the inactive state (Toyoda *et al* 2018) and six structures in an active state coupled to G proteins and bound to agonists (Nojima *et al* 2020, Huang *et al* 2024). By contrast, there were only three previously published structures of the EP2 receptor, all in the G protein coupled state and bound to agonist (Qu *et al* 2021). There are thus two aspects of the current manuscript that make it interesting. Firstly, the inactive state of the EP2 receptor, which allows a refinement of the activation mechanism (although this is essentially identical to the EP4 receptor previously published). Secondly, and more interestingly, the structures of the EP2 and EP4 receptors in the inactive state and bound to the identical antagonist, TG6-129. This represents a particularly striking example of a 'two-warhead' ligand, where one end binds and inhibits EP2, whilst the other end binds and inhibits EP4. This property of TG6-129 appears to be totally serendipitous, as the original paper describing its synthesis and characterisation was aiming only for an EP2-specific antagonist (Ganesh *et al* 2013).

The manuscript, on the whole, is reasonably clear and well written, amply supported by the figures. The first part of the results section focusses on describing the selective antagonist structures, with the

final half describing the dual inhibitor structures. As this section is the crux of the whole paper, it needs some more work to make the comparisons between the structures clearer (see below).

Major points

1. Figure 5 shows the structures of EP2 bound to TG6-129 and EP4 bound to TG6-129, but there are no panels that show direct comparisons between the two receptors bound to TG6-129 or a comparison of which residues bind the ligand. A key panel to add is an overlay of the two receptors and then highlighting (in sticks) the different orientations and positions of TG6-129 in the different receptors. Another comparison that needs to be made is to have one panel that shows side-by-side the amino acid residues interacting with TG6-129 and which helices they are in.

Response: We appreciate the referee's valuable suggestion for including direct structural comparisons that would clearly illustrate the distinct binding modes of TG6-129 in EP2 versus EP4. Such comparative visualizations are indeed essential for understanding the molecular basis of this dual antagonist's differential binding mechanisms.

To address this request, we have performed comprehensive structural overlays of TG6-129-bound EP2 and TG6-129-bound EP4 structures, which reveal striking differences in both receptor conformations and ligand positioning. The overlay clearly demonstrates how TG6-129 adopts markedly different orientations and binding depths in the two receptors, with deep insertion into the EP2 pocket versus shallow surface binding in EP4.

Additionally, we have prepared detailed side-by-side comparisons showing the specific amino acid residues that interact with each region (A, B, and C) of TG6-129 in both receptors, along with their corresponding transmembrane helix assignments. This analysis highlights the distinct interaction networks that enable TG6-129's dual antagonistic activity, with different sets of residues from different helical regions engaged in each receptor.

These structural comparisons, including the receptor overlay with TG6-129 in different orientations and the detailed residue interaction maps, are presented in **Figure EV3E**. The conformational differences between the binding pockets and the distinct amino acid interaction patterns are thoroughly described in the Results section (**pages 14, lines 346-353**). These comparisons provide crucial insights into the "two-warhead" mechanism underlying TG6-129's dual antagonistic properties.

2. It would also be helpful to show overlays of other combinations of ligands after alignment of receptors e.g. TG6-129 and grapiprant and PF-04418948.

Response: We appreciate the referee's valuable suggestion to include additional ligand overlays that would provide comprehensive comparisons of binding modes across different antagonist-receptor combinations. Such systematic comparisons are valuable for understanding the structural determinants of ligand selectivity and binding mechanisms.

To address this request, we have performed extensive structural alignments and ligand overlays for multiple combinations:

EP2 comparisons: We conducted structural alignments of PF-04418948-bound and TG6-129-bound

EP2 structures, both compared with the active EP2 structure. These overlays revealed distinct differences in transmembrane helix displacement, key residue positioning and conformations, and ligand superposition within the binding pocket, providing insights into how different antagonists stabilize the inactive state (**Figure EV4A-C**).

EP4 comparisons: We performed structural alignment of grapiprant-bound and TG6-129-bound EP4 structures, which demonstrated remarkably similar overall receptor conformations despite the different ligands. This finding suggests that EP4 maintains a consistent inactive conformation regardless of which antagonist is bound (**Figure EV4D,E**).

These systematic ligand overlays illuminate important principles of antagonist binding: while EP2 shows notable conformational differences depending on the bound antagonist, EP4 appears to adopt a more rigid inactive conformation that accommodates different ligands through local pocket adjustments rather than global conformational changes. These comparative analyses are described in detail in the main text (**see new paragraph on page 14-15, lines 358-373**) and provide valuable insights into the structural flexibility differences between EP2 and EP4.

3. The apparent resolutions quoted for the cryo-EM structures need to be for just the receptor portion of the complex as well as for the whole complex. This is important so that the reader gets an accurate description of the resolution of the receptor that is not over-inflated by the well-ordered regions of the bound antibody (which sometimes happens). Two figures therefore need to be quoted in Table S1 (map resolution of the complex and map resolution of the receptor). In addition, the local resolution figures (Sup Figs 2 and 3) need to be scaled appropriately so that the density shows a range of colour and not just all blue, which is highly uninformative. There needs to be roughly equal amounts of dark blue and dark red in each figure.

Response: We appreciate the referee's valuable suggestion regarding the accurate representation of cryo-EM resolution data. This is indeed a critical point for proper assessment of structural quality, as antibody regions can artificially inflate overall resolution estimates and potentially mislead readers about the actual quality of the receptor density.

To address this important concern, we have made the following revisions:

Resolution reporting: We have calculated and now report separate resolution values for both the entire complex and the receptor portion specifically in **Appendix Table S1**. This dual reporting approach ensures that readers can accurately assess the quality of the receptor structure independent of the well-ordered antibody components. The receptor-specific resolution values provide a more realistic assessment of the density quality for the pharmacologically relevant portions of the structures.

Local resolution visualization: We have rescaled the local resolution figures (**Appendix Figure S2-S5**) to display an appropriate dynamic range with balanced color distribution. The revised figures now show meaningful color gradients from dark blue (highest resolution) to dark red (lower resolution), with roughly equal representation across the color spectrum. This improved visualization allows readers to properly assess the local resolution variations throughout the structures and identify regions of highest and lowest density quality.

These revisions provide a more transparent and accurate representation of our structural data quality, enabling readers to make informed assessments of the reliability of different regions within our cryo-EM structures.

4. The different binding mode of TG6-129 to EP2 and EP4 is the key result in this manuscript, yet despite the claim in the manuscript of 'extensive' mutagenesis studies, this binding mode has not been validated according to the authors own criteria (as depicted in Figures 3d and 4e). For consistency and completeness, the authors should mutate all the residues in the binding pocket of EP4 and EP2 (as defined in Figs 5c and 5i) to alanine and test binding of TG6-129. It will be important to include residues in the base of the pocket of EP4 to show that the ligand does not access deeper regions where other antagonists can bind.

Response: We appreciate the referee's valuable suggestion and agree that comprehensive mutagenesis validation is essential to support our claims about TG6-129's distinct binding modes in EP2 versus EP4. The differential binding mechanism is indeed a key finding of our manuscript, and it requires thorough experimental validation through systematic mutagenesis studies.

EP2 validation: As noted by the referee, functional assays for the mutated residues in the EP2 binding pocket have been demonstrated in **Figure 5D, E** and described on **pages 12-13**. These studies confirm the deep binding mode of TG6-129 in EP2 through interactions with residues throughout the pocket depth.

EP4 validation: For consistency and completeness, we conducted systematic alanine mutagenesis of all residues in the EP4 binding pocket as defined in **Figure 5I**. We tested the antagonistic activity of TG6-129 against these mutants using the GloSensor cAMP assay. However, many mutations resulted in severely compromised receptor function (less than 20% E_{max} compared to wild-type receptor or complete loss of PGE₂-induced activation), preventing reliable assessment of antagonist potency. We successfully analyzed TG6-129 antagonism in EP4 mutants that retained sufficient function ($E_{max} > 20\%$): V72^{2.57}A, I317^{7.41}A, S319^{7.43}A and V320^{7.44}A. Importantly, mutations of V72^{2.57}A and S319^{7.43}A significantly reduced TG6-129 antagonistic potency, confirming these residues' critical roles in the shallow binding mode.

These comprehensive mutagenesis results are presented in **Figure 5D, E, K and EV3B, D** and described in the text (**pages 12-13, lines 214-322, 343-344**).

5. The methods section for the purification of the receptors is too brief and incomplete for any competent researcher to follow and repeat, so this needs to be expanded for both receptors to include full buffer compositions, temperatures, times of centrifugation etc. For example, it is unacceptable to write 'Throughout this process, a specialized buffer containing LMNG, GDN, digitonin, and CHS was utilized to maintain protein stability and ligand binding'. It is a requirement for publication that full methods are given.

Response: We appreciate the referee's valuable feedback and sincerely apologize for the inadequate detail in our original methods section. We fully agree that reproducibility is a cornerstone of scientific research, and detailed protocols are essential for other researchers to replicate our work successfully.

To address this critical deficiency, we have comprehensively revised and expanded the Methods section to include complete experimental details for both EP2 and EP4 receptor purification. The revised protocols now provide:

Complete buffer compositions: All buffer formulations are specified with exact concentrations of each component, including detergent mixtures, salt concentrations, pH values, and additive concentrations.

Detailed procedural parameters: We have included specific temperatures for each step, centrifugation speeds and durations, incubation times, column volumes for washing and elution steps, and equipment specifications.

Step-by-step protocols: The purification procedures are now described in sufficient detail to enable reproduction, including cell harvesting conditions, membrane preparation steps, solubilization parameters, affinity chromatography protocols, and size exclusion chromatography conditions.

These comprehensive methodological details are now provided in the revised Methods section, ensuring that any competent researcher can follow and reproduce our protein purification protocols. We acknowledge that this level of detail should have been included in the original submission and thank the referee for emphasizing this fundamental requirement for scientific publication.

Minor points

No page numbers or line numbers are given, so I will not correct any of the many typographical errors or areas of ambiguity in phraseology as it is too difficult to do.

Response: We appreciate the referee's valuable suggestion. To facilitate easy reading and enable precise referencing of any typographical errors or areas that need clarification, we have added page numbers and line numbers throughout the entire manuscript text.

1. Standard nomenclature for G proteins requires the subscripting of the q, s and i descriptors. Please do this throughout the text.

Response: We thank the referee for pointing out this formatting error. We have corrected the G protein nomenclature to use proper subscripting (G_q , G_s , and G_i) throughout the entire text to comply with standard biochemical nomenclature.

2. For the first mention of Fab001 in the results section, please quote the reference (number 24).

Response: We thank the referee for this reminder. We have added the appropriate reference citation (Toyoda *et al* 2019) for Fab001 at its first mention in the Results section (**see page 7, lines 137-138**).

3. Emory University should not be quoted for the development of TG6-129, cite the authors involved.

Response: We thank the referee's comment for the reference citation and have cited the authors (Ganesh *et al*, 2013) involved the development of TG6-129.

4. In the methods, state how the mutations A62L and G106R were chosen and what is their purpose.

Response: We appreciate the referee's valuable suggestion to clarify the rationale for these specific

mutations. According to previously established protocols (Toyoda *et al* 2019), the construct incorporated two strategic point mutations (Ala62^{2,47}Leu and Gly106^{3,39}Arg) specifically chosen to achieve substantially higher receptor stability in the inactive state, which is essential for successful structural studies. This explanation has been added to the Methods section (**page 21, lines 556-559**).

5. All the figures are too small. The minimum text size should be 5 font (non-serif). In many instances it is impossible to read the text because it is so small and pixellated (e.g. the FSC curves in Fig S2 and S3).

Response: We appreciate the referee's valuable suggestion regarding figure readability. Following the EMBO Journal Figure Guidelines, we have systematically revised all figures to ensure minimum text size of 7-point non-serif font and improved overall legibility. For figures containing extensive information that remained difficult to read even after resizing, we have split them into separate panels: the original Supplementary Figure 2 has been divided into **Appendix Figures S2 and S3**, and the original Supplementary Figure 3 has been split into **Appendix Figures S4 and S5**.

6. In all the legends to structure figures, colours are cited according to their names in Chimera/ChimeraX and are often very unhelpful (what colour is peru?). Please use standard colour names e.g blue, green, turquoise and define as either light or dark.

Response: We appreciate the referee's valuable suggestion regarding color clarity in figure legends. We have revised all structural figure legends to use standard, descriptive color names (e.g., light blue, light pink, light green), eliminated software-specific color terminology that may be unclear to readers unfamiliar with molecular visualization programs and carefully labeled the ligands and receptors referred to by each color in all structural figures.

7. For all graphs showing the activity of mutants, make a separate Table in supplementary giving the average values plotted, the errors, the number of replicates, the actual P values, and (where appropriate) the IC50s, EC50s and Emax values. The number of biological replicates are cited, but how many technical replicates were included in each measurement; please state in every case.

Response: We appreciate the referee's valuable suggestion for comprehensive data reporting. We have created **Appendix Table S2-S5**, which provide detailed quantitative data for all mutagenesis experiments, including mean values, standard errors, actual p-values, and pharmacological parameters (pIC₅₀, E_{max} values). Additionally, we have clarified that all functional assay data represent means \pm S.E.M. from 3 independent biological experiments, each performed with 3 technical replicates, and this information is now explicitly stated in every relevant figure legend.

8. Fig 1e legend: The colours associated with the hydrophobic and polar charged residues are mislabelled. Histidine should be included as a polar charged residue (pka ~6.5, buffers used pH 7.4). so about ~90% is charged.

Response: We thank the referee for this important biochemical point. We have carefully reviewed the color coding in **Figure 3F** (previously 3E) and confirm that histidine is correctly classified and colored as a polar charged residue, consistent with its predominantly charged state (~90%) at physiological pH (7.4) given its pKa of approximately 6.5.

9. Wild type data needs to be included in the bar graphs in Fig 3d and 4e to show data points and

errors.

Response: We appreciate the referee's valuable suggestion for complete data presentation. We have added wild-type control data with error bars to all relevant bar graphs (**Figures 3E,4E,5E,5K, and EV1E,2D,3B,3D**) to provide proper baseline comparisons and enable accurate assessment of mutational effects relative to the native receptor responses.

References

Qu C, Mao C, Xiao P, Shen Q, Zhong YN, Yang F, Shen DD, Tao X, Zhang H, Yan X *et al* (2021)

Ligand recognition, unconventional activation, and G protein coupling of the prostaglandin E(2)

receptor EP2 subtype. *Sci Adv* 7: eabf1268

Toyoda Y, Morimoto K, Suno R, Horita S, Yamashita K, Hirata K, Sekiguchi Y, Yasuda S, Shiroishi M,

Shimizu T *et al* (2019) Ligand binding to human prostaglandin E receptor EP(4) at the lipid-bilayer

interface. *Nat Chem Biol* 15: 18-26

Dear Dr. Xu,

Thank you for submitting a revised version of your manuscript. We have now received input from two of the original reviewers. While reviewer #1 are satisfied with the revisions, reviewer #3 indicates several concerns that will need to be addressed in the final revision round.

Additionally, there remain a few editorial points that need to be addressed before I can extend official acceptance of the manuscript:

1. Please submit up to five keywords for your manuscript.
2. Please check that the funding information is correct and identical both in the manuscript and our online system. Please remove the information entered into the comments section in our system and add the complete funding information to the line-by-line entries: these will be directly linked from the published article and have to be complete.
3. Please correct the order of the manuscript sections to: Abstract, Introduction, Results, Discussion, Methods, Acknowledgements, Disclosure and competing interests statement, References, Figure legends, Expanded View Figure legends.
4. We are missing the ORCID iD for the main corresponding author (yourself). In order to link the ORCID iD to the account in our manuscript tracking system, the author in question has to do the following:
 - Click the 'Modify Profile' link at the bottom of your homepage in our system.
 - On the next page you will see a box halfway down the page titled ORCID*. Below this box is red text reading 'To Register/Link to ORCID, click here'. Please follow that link: you will be taken to ORCID where you can log in to your account (or create an account if you don't have one)
 - You will then be asked to authorise Wiley to access your ORCID information. Once you have approved the linking, you will be brought back to our manuscript system.Unfortunately, we cannot do this linking on the author's behalf for security reasons.
5. CRediT has replaced the traditional author contributions section because it offers a systematic, machine-readable author contributions format that allows for more effective research assessment. Please remove the Authors Contributions from the manuscript and use the free text boxes beneath each contributing author's name in our online submission system to add specific details on the author's contribution. More information is available in our guide to authors.
6. In the "Data availability" section, please add resolvable links to the datasets.
7. In our standard source data check, we have noted unexplained duplicate values in the datasets for some of the figures. I have attached the corresponding files with the detected duplications labelled in colour. Please check - a brief explanation would be very helpful. We appreciate that such duplications can also occur due to used measurement or quantification procedures.
8. Our data editors have asked to provide the exact p values in the legends of figures 3E, 4E, 5E, EV1 E, EV3 B, D.
9. Papers published in The EMBO Journal are accompanied online by a 'Synopsis' to enhance discoverability of the manuscript. It consists of A) a short (1-2 sentences) summary of the findings and their significance, B) 3-4 bullet points highlighting key results and C) a synopsis image that is 550x300-600 pixels large (width x height, jpeg or png format). You can either show a model or key data in the synopsis image. Please note that the image size is rather small and that text needs to be readable at the final size.

With kind regards,

Ieva

We realize that it is difficult to revise to a specific deadline. In the interest of protecting the conceptual advance provided by the work, we recommend a revision within 3 months (9th Dec 2025). Please discuss the revision progress ahead of this time with the editor if you require more time to complete the revisions.

Referee #1:

The authors have well addressed my concerns with the inclusion of additional functional data and MD simulations. I have no major concerns regarding the scientific content of this manuscript.

Referee #3:

The authors have made many improvements to the manuscript and also addressed many of the concerns raised by the referees. However, this has now raised an additional issue and also some of the changes have not been made as stated in the Reply to the Referees.

Lines 167-170: MD simulations are totally inappropriate to demonstrate that the BRIL fusion does not affect the conformational state of a receptor. Receptor activation occurs on a millisecond timescale whereas the simulations performed do not even reach in total one microsecond. In all the all-atomistic (unsteered) simulations performed on GPCRs, none have observed the transition from an inactive state to an active state. All that is being observed in the MD simulations presented in this manuscript are thermal fluctuations of the structure. As the authors have not done MD simulations of the WT receptor, no comparison can be made of the BRIL-inserted receptor and therefore no conclusion can be drawn regarding the effect of the BRIL on receptor dynamics. The only way to show that the BRIL fusion is unlikely to be having an effect on the receptor is direct comparison of binding affinities of the receptor fusion with the wild-type receptor. If there is no difference in the affinities of agonists and antagonists, then this supports the contention that the BRIL has not biased the state of the receptor towards any particular conformation. If antagonist affinity is higher than WT and agonist affinity is lower, then the BRIL has preferentially stabilized the inactive state, whilst the converse suggests that the BRIL has biased the receptor towards an active-like state. This is exactly what has been observed in pharmacological measurements of ligand affinity for the beta2 receptor bound to intracellular nanobodies that stabilize specific states of the receptor, and also during the thermostabilisation by mutagenesis of the beta1 receptor and adenosine A2a receptors.

The MD simulations should be removed and the pharmacological comparison between EP2-BRIL and WT EP2 performed.

Local resolution visualization: We have rescaled the local resolution figures (Appendix Figure S2-S5) to display an appropriate dynamic range with balanced color distribution. The revised figures now show meaningful color gradients from dark blue (highest resolution) to dark red (lower resolution), with roughly equal representation across the color spectrum. This improved visualization allows readers to properly assess the local resolution variations throughout the structures and identify regions of highest and lowest density quality.

This has not been done exactly as stated. The amount of blue in the six local resolution maps presented does not equal the amount of red.

Response: We thank the referee for this important biochemical point. We have carefully reviewed the color coding in Figure 3F (previously 3E) and confirm that histidine is correctly classified and colored as a polar charged residue, consistent with its predominantly charged state (~90%) at physiological pH (7.4) given its pKa of approximately 6.5.

This has not been done. Histidine residues are coloured green and the legend still states that green residues are 'polar charged' and that blue residues are 'polar uncharged'. I would have thought it self-evident that this should also be changed in Figure 4D as well.

The additional text added to the manuscript needs correcting into grammatical English and typos corrected e.g. 'cyro' to 'cryo', 'meddle' to 'middle' etc

Manuscript ID: EMBOJ-2025-121599R

Title: Structural Insights into Selective and Dual Antagonism of EP2 and EP4 Prostaglandin Receptors

We thank the referees for their positive assessments on the quality and importance of our work. Their constructive suggestions have been invaluable in revising our manuscript.

In the following sections, we provide point-by-point responses to the comments made by the two referees of our revised paper. The referee's comments are presented in **black** and our responses are in **blue**.

Point-by-point responses to Referee #1

The authors have well addressed my concerns with the inclusion of additional functional data and MD simulations. I have no major concerns regarding the scientific content of this manuscript.

Response: We sincerely and gratefully appreciate the referee's positive feedback and recognition of our revisions. This feedback not only validates the efforts we invested in strengthening the study but also reinforces our confidence in the robustness of our scientific content.

Point-by-point responses to Referee #3

The authors have made many improvements to the manuscript and also addressed many of the concerns raised by the referees. However, this has now raised an additional issue and also some of the changes have not been made as stated in the Reply to the Referees.

Lines 167-170: MD simulations are totally inappropriate to demonstrate that the BRIL fusion does not affect the conformational state of a receptor. Receptor activation occurs on a millisecond timescale whereas the simulations performed do not even reach in total one microsecond. In all the all-atomistic (unsteered) simulations performed on GPCRs, none have observed the transition from an inactive state to an active state. All that is being observed in the MD simulations presented in this manuscript are thermal fluctuations of the structure. As the authors have not done MD simulations of the WT receptor, no comparison can be made of the BRIL-inserted receptor and therefore no conclusion can be drawn regarding the effect of the BRIL on receptor dynamics.

The only way to show that the BRIL fusion is unlikely to be having an effect on the receptor is direct comparison of binding affinities of the receptor fusion with the wild-type receptor. If there is no difference in the affinities of agonists and antagonists, then this supports the contention that the BRIL has not biased the state of the receptor towards any particular conformation. If antagonist affinity is higher than WT and agonist affinity is lower, then the BRIL has preferentially stabilized the inactive state, whilst the converse suggests that the BRIL has biased the receptor towards an active-like state. This is exactly what has been observed in pharmacological measurements of ligand affinity for the beta2 receptor bound to intracellular nanobodies that stabilize specific states of the receptor, and also during the thermostabilisation by mutagenesis of the beta1 receptor and adenosine A2a receptors.

The MD simulations should be removed and the pharmacological comparison between EP2-BRIL and WT EP2 performed.

Response: We thank the referee for pointing out this important methodological issue. We acknowledge that our MD simulations cannot adequately support conclusions about the effects of BRIL fusion on receptor conformational states. We completely agree with the referee that the timescale limitations of our simulations and the absence of direct comparison with wild-type receptor make it inappropriate to draw definitive conclusions about conformational neutrality from these data.

In response to this valid criticism, we have weakened and modified the related descriptions in our manuscript. Specifically, we have removed the strong conclusion that our MD simulations demonstrate a "native-like inactive state", eliminated Appendix Figure S6A and associated text that overstated what our simulation data could support.

Additionally, we respectfully note that numerous studies suggest that the BRIL fusion strategy does not inherently constrain the receptor's conformational state. Both active and inactive conformations of BRIL-fused GPCRs have been successfully resolved, whether in apo form or with various ligands bound. These include:

Apo state structures: unliganded human Frizzled₁ICL₃BRIL/Fab/Nb complex (Tsutsumi et al, 2020), human GPP61₁ICL₃BRIL/Fab/Nb complex (Lees et al, 2023).

Active state structures: active serotonin receptor 5HT_{1B} structure with BRIL insertion (Mukherjee et al, 2020; Wang et al, 2013).

Inactive state structures: antagonist bound human GPP61₁ICL₃BRIL/Fab/Nb complex (Lees et al, 2023).

Multiple receptors showing different conformational states with BRIL insertions depending on bound ligands rather than the fusion construct itself. However, we acknowledge that the lack of direct pharmacological comparison between our BRIL-fused and wild-type EP2 receptors represents a limitation of our study. Our revised manuscript will present a more conservative interpretation that BRIL insertions do not inherently bias GPCR conformational states (**see new paragraph on page 16-17, lines 426-433**).

Local resolution visualization: We have rescaled the local resolution figures (Appendix Figure S2-S5) to display an appropriate dynamic range with balanced color distribution. The revised figures now show meaningful color gradients from dark blue (highest resolution) to dark red (lower resolution), with roughly equal representation across the color spectrum. This improved visualization allows readers to properly assess the local resolution variations throughout the structures and identify regions of highest and lowest density quality.

This has not been done exactly as stated. The amount of blue in the six local resolution maps presented does not equal the amount of red.

Response: We appreciate the referee's helpful suggestion and apologizes for not making the necessary modifications exactly as stated. We have rescaled the local resolution figures (**Appendix Figure S2-S5**) with balanced blue (highest resolution) to red (lower resolution) to display roughly equal representation across the color spectrum.

Response: We thank the referee for this important biochemical point. We have carefully reviewed the color coding in Figure 3F (previously 3E) and confirm that histidine is correctly classified and colored as a polar charged residue, consistent with its predominantly charged state (~90%) at physiological pH (7.4) given its pKa of approximately 6.5.

This has not been done. Histidine residues are coloured green and the legend still states that green residues are 'polar charged' and that blue residues are 'polar uncharged'. I would have thought it self-evident that this should also be changed in Figure 4D as well.

Response: We appreciate the referee for pointing out this important biochemical error and sincerely apologize for mislabeling the colors associated with the polar charged and polar uncharged residues in Figure 3F and Figure 4D. We have classified histidine as a polar charged residue in blue, and corrected the legends in **Figure 3F** and **Figure 4D** as "polar charged residues in blue, and polar uncharged residues in green".

The additional text added to the manuscript needs correcting into grammatical English and typos corrected e.g. 'cyro' to 'cryo', 'meddle' to 'middle' etc

Response: We are very sorry for our negligence, and thank the referee for the careful review and for identifying these typographical errors. We have corrected the misspelling of "cryo" and "middle" on the figure legends and have conducted a comprehensive proofreading of the entire manuscript to identify and correct any additional typographical errors or inconsistencies.

References

- Lees JA, Dias JM, Rajamohan F, Fortin JP, O'Connor R, Kong JX, Hughes EAG, Fisher EL, Tuttle JB, Lovett G *et al* (2023) An inverse agonist of orphan receptor GPR61 acts by a G protein-competitive allosteric mechanism. *Nat Commun* 14: 5938
- Mukherjee S, Erramilli SK, Ammirati M, Alvarez FJD, Fennell KF, Purdy MD, Skrobek BM, Radziwon K, Coukos J, Kang Y *et al* (2020) Synthetic antibodies against BRIL as universal fiducial marks for single-particle cryoEM structure determination of membrane proteins. *Nat Commun* 11: 1598
- Tsutsumi N, Mukherjee S, Waghay D, Janda CY, Jude KM, Miao Y, Burg JS, Aduri NG, Kossiakoff AA, Gati C *et al* (2020) Structure of human Frizzled5 by fiducial-assisted cryo-EM supports a heterodimeric mechanism of canonical Wnt signaling. *Elife* 9: e58464
- Wang C, Jiang Y, Ma J, Wu H, Wacker D, Katritch V, Han GW, Liu W, Huang XP, Vardy E *et al* (2013) Structural basis for molecular recognition at serotonin receptors. *Science* 340: 610-614

Dear Dr. Xu,

Thank you for incorporating the final changes in the manuscript. I am now pleased to inform you that your manuscript has been accepted for publication.

Before we forward your manuscript to the publishers, some modifications in the synopsis image would be needed. Please correct the text to "micelle" (instead of "micell") and "'dual warhead" interaction' instead of "manner". Please also consider slightly increasing the font size, as the text becomes difficult to discern when the image is reformatted to 550 pixels in width according to our technical requirements. You can send me the updated file via email.

I will also look into the synopsis text that you kindly provided and will let you know by Monday if any edits to the journal style are needed.

If you have any questions, please do not hesitate to contact the Editorial Office. Thank you for this contribution to The EMBO Journal and congratulations on a nice study!

Best wishes,

leva

leva Gailite, PhD
Senior Scientific Editor
The EMBO Journal
Meyerhofstrasse 1
D-69117 Heidelberg
Tel: +4962218891309
i.gailite@embojournal.org
